# Knowledge integration and decision support for accelerated discovery of antibiotic resistance genes

Jason Youn[1,2,3], Navneet Rai[1,2,3] & Ilias Tagkopoulos[1,2,3 ✉]

We present a machine learning framework to automate knowledge discovery through knowledge graph construction, inconsistency resolution, and iterative link prediction. By incorporating knowledge from 10 publicly available sources, we construct an *Escherichia coli* antibiotic resistance knowledge graph with 651,758 triples from 23 triple types after resolving 236 sets of inconsistencies. Iteratively applying link prediction to this graph and wet-lab validation of the generated hypotheses reveal 15 antibiotic resistant *E. coli* genes, with 6 of them never associated with antibiotic resistance for any microbe. Iterative link prediction leads to a performance improvement and more findings. The probability of positive findings highly correlates with experimentally validated findings ($R^2 = 0.94$). We also identify 5 homologs in *Salmonella enterica* that are all validated to confer resistance to antibiotics. This work demonstrates how evidence-driven decisions are a step toward automating knowledge discovery with high confidence and accelerated pace, thereby substituting traditional time-consuming and expensive methods.

[1] Department of Computer Science, University of California, Davis, CA 95616, USA. [2] Genome Center, University of California, Davis, CA 95616, USA. [3] USDA/NSF AI Institute for Next Generation Food Systems (AIFS), University of California, Davis, CA 95616, USA. ✉email: itagkopoulos@ucdavis.edu

For computational methods to be effective, the integration and ingestion of biological data at scale are paramount[1–3]. To this end, various initiatives[4–6] have transitioned from relational databases that store data using tables and are often limited by their scalability[7] to graph databases that efficiently process dense interrelated datasets[8] by utilizing the Resource Description Framework (RDF) triple of subject, predicate, and object[9]. This design helps to identify patterns among data, and to utilize the information content they carry to gain insights into the mechanisms of action, associations, and testable hypotheses[2,10]. In the biomedical domain, knowledge graphs[11] with thousands to millions of RDF triples are used to organize knowledge in life sciences[12,13], including health conditions such as cancer[14] and cardiovascular disease[15]. In the case of antibiotic resistance genes (ARGs), there exist both graph databases like CARD[16] and ARDB[17] that represent ontologies, as well as traditional databases like MEGARes[18], ARGO[19], and ARG-ANNOT[20] that store ARG sequencing data. Current challenges include unreported or unresolved conflicted information between two or more sources[21,22], lack of negative findings[23,24] that is necessary to train machine learning models, focus on only one relation type[25], inability to directly integrate results across sources due to incompatible meta-data[26], all of which limits their suitability as a training set for machine learning models. Similarly, extracting training data from published literature is challenging as it is often hidden in supplementary tables and figures[27,28], may be inaccessible or incompatible[29,30], which hinders any knowledge synthesis and analysis[31,32].

Automating the integration of heterogeneous biomedical data and their organization so they are machine learning-ready for downstream analysis and knowledge discovery is important for any life science field. One such area is the discovery of ARGs and relationships. Antibiotic resistance poses a major threat to the efficacy of the antibacterial drugs, which leads to increased mortality and costs[33]. Identification of ARGs has traditionally been performed through time-consuming and expensive culture-based methods[34] and more recently through bioinformatics analysis of whole-genome sequencing samples, including BLAST-based[20,35] and deep learning-based[36,37] methods. Outside of the domain of antibiotic resistance, there have been multiple attempts to discover biological knowledge from knowledge graphs[38–42] by formulating it as a knowledge graph completion (KGC)[43] problem, where the objective is to complete (discover) the missing links (new knowledge) in the graph. Graph feature models[44,45] and latent feature models[46,47] have traditionally been used for KGC, whereas models that utilize pre-trained language models (LM)[48,49] have recently achieved state-of-the-art results.

In this study, we present a methodology (Knowledge Integration and Decision Support, or KIDS) that constructs an inconsistency-free knowledge graph that supports multiple triple types and can be used to generate hypotheses over multiple iterations (Fig. 1). We apply the KIDS framework to the area of *Escherichia coli* antibiotic resistance, which leads to a knowledge graph consisting of 651,758 triples of 23 RDF triple types in total, among which 9 triple types are negative. To resolve inconsistencies, we computationally predicted, and experimentally validated 236 sets of inconsistencies with 94.07% accuracy. We then demonstrate how the automated process allows the discovery of previously unknown ARGs. KIDS achieved an average of 0.77 AUCPR and 0.86 AUROC in predicting the ARGs over two iterations of hypothesis generation, validation, and integration with existing knowledge, with the predicted ARG probability being highly correlated with validated findings ($R^2 = 0.94$). Furthermore, our analysis led to the discovery of six ARGs that we have validated experimentally, among which five homologs in *Salmonella enterica* also showed antibiotic resistance.

## Results

**The landscape of *E. coli* antibiotic resistance genes and processes**. We applied the KIDS framework on the biological domain of *E. coli* and constructed a multi-relational knowledge graph[50] (see Methods) that consists of 651,758 triples (Fig. 2a, b and Supplementary Data 1). Raw data to construct the knowledge graph were curated from a total of ten sources (Section 1.1.1 of Supplementary Information) that include information about antibiotic resistance, effects of antibiotics on the expression patterns, gene-regulatory relations with transcription factors, and the impact of genes on the biology of an organism at the molecular, cellular, and organism levels[51], all regarding *E. coli* genes (Fig. 2c). The resulting knowledge graph provides a comprehensive view of the positive *E. coli* antibiotic resistance with 18-fold more genes and 3-fold more antibiotics than CARD[16] (Fig. 2d, Supplementary Table 1). Among the 23 triple types of the knowledge graph, 14 positive triple types account for the 31,216 (4.8%) associations as genes are less likely to confer resistance to an antibiotic (Fig. 2e). The knowledge graph contains antibiotic exposure times at six different time points ranging from 30 min to 7 days (Supplementary Table 2). From the total of 466,752 possible gene-antibiotic pairs, 358,674 pairs (76.9%) were connected via either a positive or negative 'confers resistance to antibiotic (CRA)' predicate, with the rest being candidates for either association (Supplementary Fig. 1).

**Resolved inconsistencies help discover new knowledge**. We identified 236 sets of inconsistencies in our intermediate knowledge graph (Supplementary Data 2, Fig. 3a) between the findings of two sources Tamae et al.[21] and Liu et al.[52] for positive and negative counterparts of the predicate 'CRA after 18 h' despite their identical experimental setup (Supplementary Table 6). We then applied the AverageLog[53] inconsistency resolution algorithm (see Methods) to select which one of the two conflicting facts is more likely to be true by iteratively updating the source trustworthiness and belief of triple (Fig. 3b). Results show that we were able to accurately resolve these inconsistencies (94.07% accuracy, 50.0% F1-score, 33.3% precision, 3.0% baseline precision) when compared to the ground truth wet-lab validation (Fig. 3b, Supplementary Table 3), which was performed by measuring and comparing the minimum inhibitory concentrations (MICs) of the single-gene knock-out strain and the wild-type strain on the LB agar plate (see Methods, Supplementary Data 8). We then trained the hypothesis generator before and after resolving inconsistencies, to test how inconsistency resolution affects knowledge discovery. This led to two previously unidentified antibiotic-resistant relationships (*surA*, CRA, Vancomycin) and (*asmA*, CRA, Vancomycin) with significantly increased probabilities after the inconsistency resolution (0.024–0.882 and 0.005–0.213, respectively) that we validated experimentally.

**KIDS accelerates knowledge discovery**. The hypothesis generator module performs link prediction[43] on the incomplete knowledge graph to identify the missing links (i.e., generate hypotheses). We focused on exploring the missing CRA links between all pairwise combinations of *E. coli* genes and antibiotics (108,078 hypotheses). To this end, we applied five different variations of the hypothesis generation methods (PRA[44,54], MLP, a stacked model that combines PRA and MLP using AdaBoost[55], TransE[46], and TransD[56]; see Fig. 4b and Methods) on a reduced knowledge graph without temporal information (see Methods) that has 494,819 triples and 12 predicate types (Supplementary Table 4, Supplementary Data 1). From those methods, PRA[44,54] finds observable predicate paths between subject (source) and object (target) nodes in the graph and treats them as human-interpretable features (Supplementary

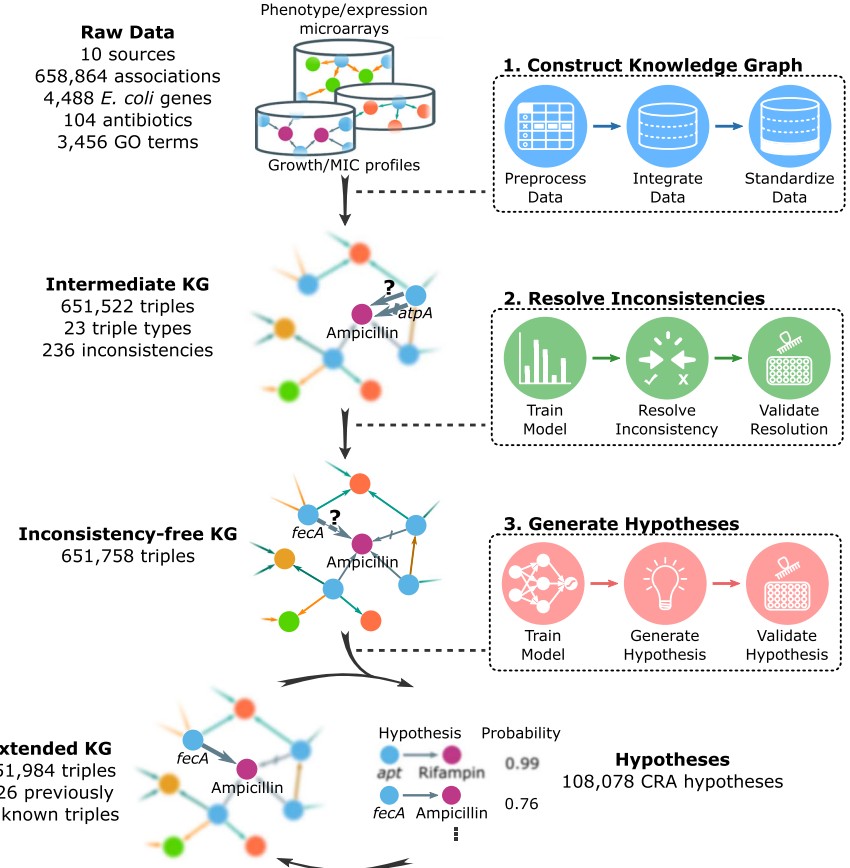

**Fig. 1 Overview of the KIDS framework.** First, an intermediate knowledge graph is created from ten sources by processing RDF triples that encode 23 types of associations. Second, inconsistencies are computationally resolved and experimentally validated to construct an inconsistency-free knowledge graph. Third, a hypothesis generator is trained on the knowledge graph and assigns probabilities for the missing links. Hypotheses with high probability are experimentally validated, and the results are integrated into the knowledge graph, which is used for the next iteration of hypothesis generation. GO refers to the Gene Ontology, and gray arrows denote the positive predicate 'confers resistance to antibiotic (CRA)'.

Table 9). In contrast, MLP[57] is a fully connected neural network that uses the triples represented by latent vector embeddings to predict whether any given edge is valid. We also tested translation-based graph embedding methods TransE[46] and TransD[56] (Section 1.3.5 of Supplementary Information), but we selected the stacked model as it had superior performance in testing. Evaluation of these methods, that have been optimized for F1-score, using 5-fold cross-validation shows that the stacked model had the best performance in terms of AUCPR with a 154.4% increase when compared to PRA (0.28 vs. 0.11, respectively, $p$ value $= 2.1 \times 10^{-6}$) and a 27.7% increase when compared to MLP (0.28 vs. 0.22, respectively, $p$ value $= 3.0 \times 10^{-3}$) (Fig. 4c, Supplementary Table 5), while the baseline was 0.02.

We used the stacked model to generate 226 CRA hypotheses of varying probability that we subsequently tested experimentally (Supplementary Data 3). Of those hypotheses, 64 (28.3%) were validated as positives (Fig. 5a). After adding those results to the knowledge graph, we ran a second iteration of KIDS, which produced another 90 hypotheses, from which 29 (28.8%) were positively validated (Fig. 5a). From these two iterations, we computationally predicted and experimentally validated, similar to the wet-lab validation performed for the inconsistency resolver (Section 1.3.12 of Supplementary Information), a total of 93 CRA hypotheses for 83 E. coli genes that confer resistance to one or more of 15 antibiotics (Fig. 5e, Supplementary Data 4). The KIDS-generated hypotheses are reliable as the calibrated output for each hypothesis is a highly correlated confidence score ($R^2 = 0.94$) with the validated outcome (Fig. 5a). For instance, hypotheses with

probability >0.8 have a high true positive rate with 29 out of 37 tested hypotheses (78.4%) to yield an ARG, whereas hypotheses with probability ≤0.2 have a true positive rate with only 14 out of 163 tested hypotheses (8.59%) to yield an ARG. Interestingly, KIDS produced improved hypotheses in the second iteration with the addition of the newly discovered results (Fig. 5b–d). The KIDS-generated hypotheses are positively correlated with high consistency when compared to the random baseline (Kendall's tau[58] = 0.96 vs. 0.00, respectively, $p$ value $< 2.2 \times 10^{-308}$; RBO[59] = 0.56 vs. 0.00, respectively, $p$ value $< 2.2 \times 10^{-308}$; Section 1.3.11 of Supplementary Information).

**AI-driven discovery of six antibiotic resistance genes**. Extensive literature search on the 83 E. coli genes that are implicated in the CRA hypotheses identified 15 genes that are previously unknown ARG for E. coli, with 6 of them (1 from the first iteration and 5 from the second iteration) not appearing as an ARG for any bacteria. Those 6 are the following: *ftsP*, *hdfR*, *lrp*, *proV*, *qorB*, and *rbsK* (Fig. 6), which have never been reported to be involved in antibiotic resistance (Supplementary Data 4). Further investigation on the biological processes reveals they are part of a diverse repertoire of functions related to amino acid metabolism, nutrient transport, and regulation. More specifically, FtsP is a cell division protein that is required for bacterial growth during stress conditions. FtsP stabilizes or protects the divisional assembly during stress condition[60]. HdfR, which is an H-NS-dependent *flhDC* regulator, represses the expression of the flagellar master operon

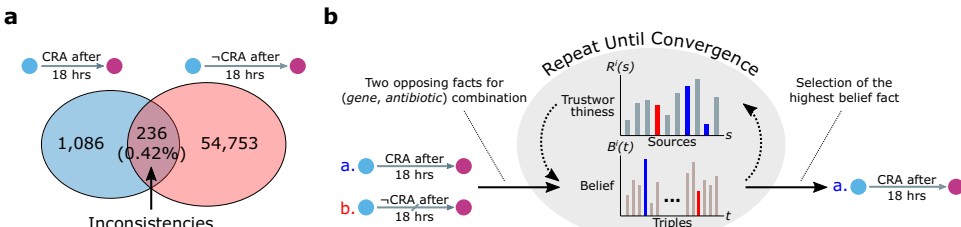

**Fig. 2 The inconsistency-free *E. coli* knowledge graph. a** Hive plot visualization of the knowledge graph's major components, with each axis corresponding to one of five different node types: gene, antibiotic, cellular component, biological process, and molecular function. The size of a node is its in and out degree. Only the 5% highest degree nodes from each node type and their positive connections are shown. **b** The top highest degree nodes for each of the eight positive predicates in the knowledge graph. **c–e** Breakdown of the knowledge graph representation in terms of data sources, node, and predicate types. CRA denotes the predicate 'confers resistance to antibiotic', whereas ¬CRA denotes 'confers no resistance to antibiotic'.

**Fig. 3 Inconsistency resolution. a** Venn diagram showing the inconsistencies detected in the intermediate knowledge graph, where the inconsistency is defined as two or more sources supporting a conflicting fact. **b** The inconsistency resolution algorithm is iteratively trained using the intermediate knowledge graph. Once the training converges, it is used to select the triple with the higher belief among the inconsistencies. See Methods for more details. The blue and purple nodes represent gene and antibiotic, respectively. CRA denotes the predicate 'confers resistance to antibiotic', whereas ¬CRA denotes 'confers no resistance to antibiotic'.

*flhDC*[61] and induces the expression of the *gltBD* operon, which is involved in acid resistance[62,63]. Lrp encodes a leucine-responsive regulatory protein, which regulates at least 10% of the genes in *E.*

*coli*, including regulation of major porins OmpC and OmpF that determine the permeability of the cell membrane[64,65]. ProV is predicted to be a component of an osmoresponsive ABC

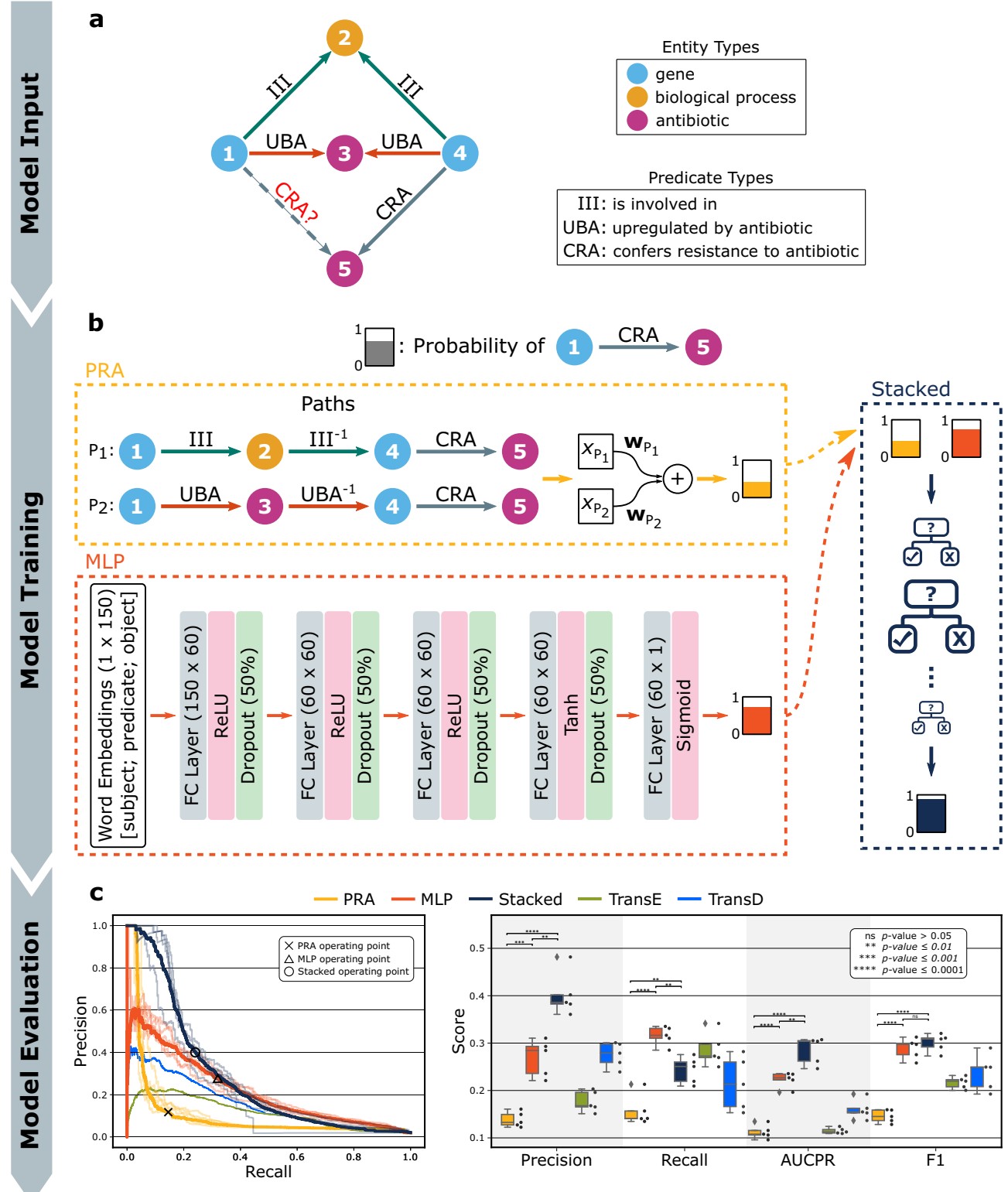

transport system and involved in osmosensing[66]. QorB is a NAD(P)H:quinone oxidoreductase, which catalyzes the reduction of quinone. *E. coli* strain overexpressing *qorB* shows defects in growth and a significant decrease in several enzymes involved in carbon metabolism[67]. Interestingly, oxidoreductases have been reported to involve antibiotic resistance[68]. RbsK is a sugar kinase that, in addition to phosphorylation of ribose, facilitates stress-induced mutagenesis in *E. coli*[69]. Mutations in sugar kinase genes

such as *waaP* of *S. enterica* lead to increase susceptibility to antibiotic polymyxin[70,71].

We did not identify any statistically significant homologs (*E* value < 0.05) of these six genes among the 4577 ARGs from CARD[72], while the best hit was for *OXA-541* of *Pseudomonas putida* for *lrp* (91.7% sequence similarity, *E* value = 0.12) (Section 1.3.13 of Supplementary Information, Supplementary Data 10). The prevalence of these six genes across the human digestive

**Fig. 4 Hypothesis generator architecture, training, and evaluation. a** Illustration of the training and evaluation of the hypothesis generator (HG). The task of the HG is to associate a probability to a putative link for the 'confers resistance to antibiotic' or CRA between two nodes (nodes 1 and 5 here). **b** Three HG architectures, PRA, MLP, and Stacked, an ensemble method of a majority voting schema of the other two, were constructed and evaluated. Additional translation-based models like TransE and TransD were also tested although not illustrated here (see Methods). **c** Precision-recall, AUCPR, and F1-score for the five methods ($n = 5$, 5-fold cross-validation). Black circles denote raw data points. The box represents the interquartile range, the middle line represents the median, the whisker line extends from minimum to maximum values, and the diamond represents outliers. For PRA vs. MLP, all scores were statistically significant (precision $p$ value $= 1.1 \times 10^{-4}$, recall $p$ value $= 1.4 \times 10^{-5}$; AUCPR $p$ value $= 2.9 \times 10^{-6}$; F1-score $p$ value $= 1.6 \times 10^{-6}$). For PRA vs. Stacked, all scores were also statistically significant (precision $p$ value $= 2.2 \times 10^{-6}$, recall $p$ value $= 2.0 \times 10^{-3}$; AUCPR $p$ value $= 2.1 \times 10^{-6}$; F1-score $p$ value $= 3.9 \times 10^{-7}$). Finally, for MLP vs. Stacked, all scores were significant (precision $p$ value $= 1.1 \times 10^{-3}$; recall $p$ value $= 1.5 \times 10^{-3}$; AUCPR $p$ value $= 3.0 \times 10^{-3}$) except for F1-score ($p$ value $= 0.37$). Note that all methods have been optimized for the F1-score, and the $p$ values were calculated using the two-sided $t$-test.

microbiome ranges from 0.67 to 8.79% (Section 1.3.14 of Supplementary Information, Supplementary Table 12). Finally, to investigate the antibiotic resistivity of genes homologous to the six previously unknown ARGs in other bacterial genera, we identified five homologs *ftsP*, *lrp*, *proV*, *rbsK*, and *yifA* (*hdfR* in *E. coli*) in *S. enterica* with >78% similarity in nucleotide sequences, while the homolog of *qorB* was not identified (Section 1.3.15 and 1.3.16 of Supplementary Information). Wet-lab validation revealed that knocking out these five genes in *S. enterica* also increased susceptibility to antibiotics (Supplementary Data 9).

## Discussion
In this work, we presented the KIDS framework as an automated method for knowledge organization and discovery. We demonstrated the power of the KIDS platform in the context of *E. coli* antibiotic resistance, a research area with a need for such a method, as the emergence of antibiotic resistance renders existing antibacterial drugs less efficient and thus necessitates a constant race to discover new ways to fight microbial infections[73]. Out of the 6 ARGs discovered in this work, we found that 5 homologs in *S. enterica* also conferred resistance to an antibiotic, indicating that the knowledge gained in this study can be easily translated to closely related bacteria. Current computational tools identify potential ARGs by genomic and metagenomic sequence analysis, which has limited performance when the reference database does not include similar ARG sequences. Similarly, just looking at homology is not sufficient for discovering ARGs. Among the 129 genes from the lowest probability range [0.0, 0.2] that we have validated to have no antibiotic resistance, we found 9 homologous ARGs in CARD that have significant *E* value (<0.05) with >68.6% sequence similarity (Supplementary Data 10). KIDS removes the dependency to reference sequences as its power stems from guilt-by-association and pattern discovery within the knowledge graph. Although manual literature curation and experimental validation were tedious and time-consuming, we found that the KIDS framework generates actionable hypotheses that lead to automated knowledge discovery with high confidence and efficiency.

On the summary statistics, the improvement from resolved inconsistencies was small, most likely because only 7 out of the 236 inconsistencies (3.0%) we experimentally resolved and further validated in the wet-lab were positive triples (Supplementary Data 2), and therefore reinstating them back to the knowledge graph where 1606 positive CRA triples exist (Supplementary Data 1) did not affect the knowledge graph much (1606–1613, a 0.44% increase). However, we found two previously unknown antibiotic-resistant relationships (*surA*, CRA, Vancomycin) and (*asmA*, CRA, Vancomycin) only after reinstating the resolved inconsistencies into the knowledge graph, something that demonstrates the importance of inconsistency resolution and coherence in our knowledge. For the lack of negative findings, our knowledge graph is the first to include both the positive findings (14 triple types, 31,216 triples) and the negative findings (9 triple

types, 620,542 triples) to the best of our knowledge. Although the majority of the hypothesis generation models we tested did not use these negatives and instead generated them either through closed-world assumption or corruption through random sampling, our best model (stacked) did utilize these negatives. We believe there is still a potential to take advantage of these negative findings in other machine learning models. To address the focus on only one relation type, our knowledge graph contains 23 relation types (Supplementary Table 2) as opposed to a single relation type from other sources (Section 1.1.1 of Supplementary Information). Finally, regarding the inability to directly integrate results across sources due to incompatible meta-data, this is still a problem for this and any other framework, as it is related to data incompatibility during their generation and reporting, something that we as a community need to collaboratively work on by adhering to standards like FAIR[74].

Although translation-based graph embedding models have shown state-of-the-art performance in some benchmark datasets[75,76], they performed worse than models like MLP and Stacked for our *E. coli* knowledge graph (Supplementary Table 5). This may be due to the known limitations of these methods where they are unable to handle knowledge graphs with complex and diverse entities and relations (e.g., one-to-many, many-to-one, many-to-many)[77] or do not utilize semantic information[78]. For example, in our knowledge graph, many genes are known to confer resistance to a specific antibiotic (many-to-one). Therefore, these genes will be close to each other in the embedding space, making it difficult to differentiate them from each other. This leaves room for performance improvement of the hypothesis generation methods by utilizing the current state-of-the-art link prediction methods[48,49] which take advantage of pre-trained LM like BERT[28] and RoBERTa[79] and approach the problem as a natural language processing task. Unlike graph embedding approaches[46,80–83], LM-based methods generalize well to unseen nodes or edges in graph[49]. However, the application of such methods on the biological domain remains a challenge as LM models are usually not trained on biological data, except BioBERT[84], in which case further fine-tuning of the LM model to the specific domain (*E. coli* ARG here) is desired. For the scope of this work, we used a stacked (MLP and PRA) hypothesis generation method, inspired by the Knowledge Vault[57] project.

There are a few areas of improvement. First, knowledge inference rules (see Methods) were generated upon visual inspection of the 23 triple types of the knowledge graph. There are automatic knowledge graph construction methods[85,86] that can potentially do this automatically, but we leave it for future work as their precision is not at the human level nor has been tested in the biomedical domain. Second, although our knowledge graph contains temporal information, we discarded them when training the hypothesis generator. Allowing temporal features[87–89], we could expand our research to generate time-specific hypotheses, using techniques such as sequence-to-sequence learning methods[90,91]. Third, the major bottleneck of

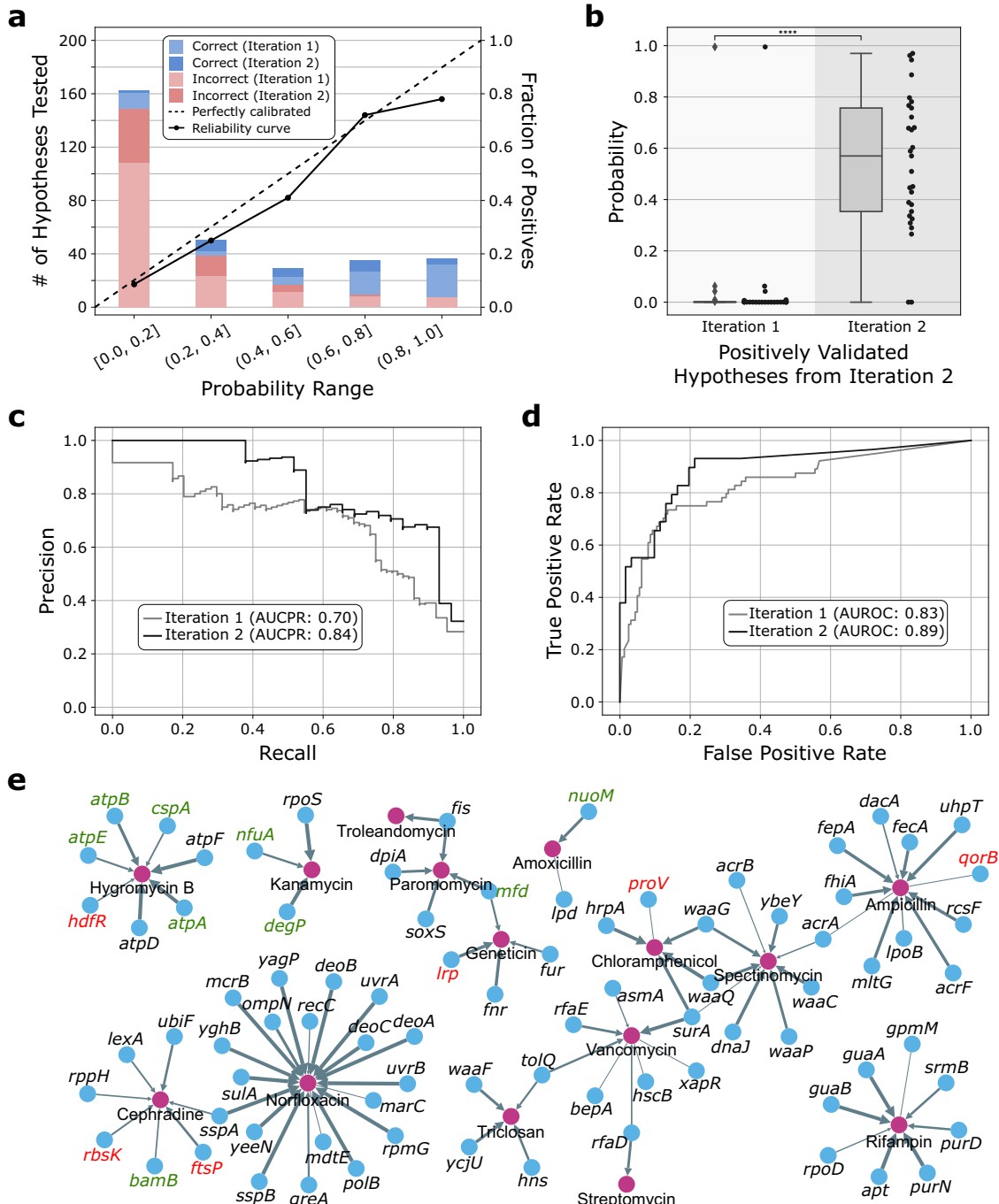

**Fig. 5 Accelerated missing link discovery through iterative learning. a** A high correlation between the probability assignment by the hypothesis generator and forward experimental validation (226 and 90 validated hypotheses from the first and second iteration, respectively; $R^2 = 0.94$). **b** The probability distribution of the positively validated hypotheses from the second iteration (i.e., dark blue bar in **a**) compared to the probability of the same hypotheses from the first iteration ($n = 29$ positively validated second iteration hypotheses). Updating the knowledge graph with the validated hypotheses in the first iteration (i.e., light blue and red bars in **a**) and re-training of the hypothesis generator led to the 14-fold probability increase (0.55 vs. 0.04, respectively, $p$ value $= 1.1 \times 10^{-11}$), which in turn enabled the discovery that would not have been possible with only one iteration of hypothesis generation. The box represents the interquartile range, the middle line represents the median, the whisker line extends from minimum to maximum values, and the diamond represents outliers. The $p$ value was calculated using the two-sided $t$-test. **c**, **d** The precision-recall (PR) and receiver operating characteristic (ROC) curves of the generated hypotheses compared against our wet-lab validation results. The AUCPR and AUROC of the second iteration hypotheses increased by 19.4% and 7.3%, respectively, when compared to the first iteration hypotheses. **e** We predicted and validated 64 CRA hypotheses from iteration 1 and 29 CRA hypotheses from iteration 2 for a total of 83 *E. coli* genes (blue node) that confer resistance (gray arrow) to one or more of 15 antibiotics (purple node). Genes with green and red labels indicate previously unknown genes that are not associated with antibiotic resistance in *E. coli* (9 genes) or any microbe (6 genes), respectively. The edge thickness is proportional to the KIDS predicted probability.

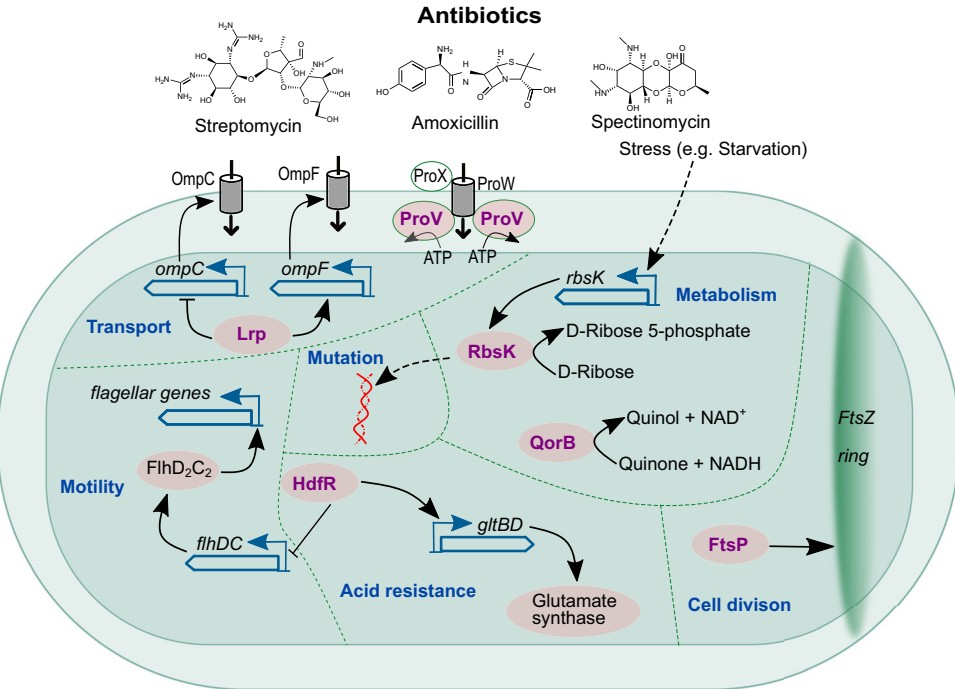

**Fig. 6 Mode of action of 6 previously unknown genes discovered to be involved in antibiotic resistance.** The proteins of these genes are shown in purple. Solid arrows indicate upregulation while blocking bars indicate downregulation. Dotted arrows indicate indirect regulation.

the KIDS framework is its dependency on expert-guided manual curation of data in RDF triple format. An automated data curator would be a boon to adding information from existing literaure[86,92]. In addition, we expect better initialization schemes, such as those based on pre-trained word embeddings trained using scientific literature instead of random initialization, to further improve performance[93–95]. Concomitantly, we would like to apply KIDS to other bacteria and replicate the success that we observe in *E. coli*. Finally, evaluating the impact of data size on learning performance (Supplementary Fig. 1) can help to determine how well this method can generalize to other microbes with limited training data.

Taken together with other advances in optimal experimental design[96,97], interpretable machine learning[98,99], and automated research and development approaches[100], the proposed framework paves the way for a systematic, optimized, and reproducible way to elucidate complex biological systems in shorter timescales, with less manual labor, and unprecedented fidelity.

## Methods

**Knowledge graph constructor**. The knowledge graph construction process is shown in Supplementary Fig. 2 with detailed examples.

*Data integration*. We merge distinctive sets of knowledge from ten different sources (Section 1.1.1 of Supplementary Information) in a unified format using binary relationships known as an RDF triple of the form (subject, predicate, object), where subject and object are the nodes (biological entities) in the graph, and the predicate is the edge (relation) between them.

*Synonym resolution*. For entity types gene and antibiotic in the integrated knowledge graph, a name mapping table is applied to resolve the synonyms as multiple representations may exist for a single entity. For gene name mapping, Accession IDs to external databases and synonym lists of all *E. coli* genes downloaded from EcoCyc[101] are mapped to the original gene symbol. For all antibiotics, we map all synonyms listed in ChemIDplus[102] to its MeSH name (defined as MeSH heading in ChemIDplus). This name mapping table is in Supplementary Data 5.

*Knowledge inference*. As a data augmentation step, we added 15 sets of rules that we manually created to bridge existing gaps in the knowledge representation. As an

example, a new triple (*sucD*, has, response to antibiotic) can be inferred from an existing triple (*sucD*, CRA, Cephradine). The full list of rules created is listed in Supplementary Data 6.

**Inconsistency resolver**. The inconsistency resolution process is shown in Supplementary Fig. 2 with detailed examples.

*Inconsistency detection*. To detect inconsistencies in the knowledge graph, we manually defined nine sets of rules (Supplementary Data 7) upon close inspection of the knowledge graph. In this work, we treat a set of triples that share the same subject and object entities connected by conflicting predicates as an inconsistency. For example, triples (*atpA*, CRA after 18 h, Ampicillin) and (*atpA*, confers no resistance to antibiotic after 18 h, Ampicillin) are considered one set of inconsistency.

*Inconsistency resolution*. Let $t$ be a triple and $s$ be a source, then $t \in T$ and $s \in S$, where $T$ and $S$ is the group of all triples and all sources, respectively. If we let $T_s$ be all the triples of source $s$, then $T = \bigcup_{s \in S} T_s$. Each triple $t \in T$ belongs to a mutual exclusion set $M_t \subseteq T$, a set of triples that are mutually exclusive with one another. In an inconsistency-free setting, a triple $t$ belongs to one unique $M_t$. In other words, $|M_t| = 1$ means there exist no conflicts in $M_t$. Assuming there exists one true triple $\bar{t}$ in each mutual exclusion set $M$, the goal of the inconsistency correction methods is to predict $\bar{t}$ for all $M$ with $1<|M|$. Prediction of $\bar{t}$ is done by measuring belief of triple $t$, $B(t)$ (i.e., the level of confidence that triple $t$ is true), among all $t$ in $M$ and by assigning $t$ with the highest belief $\text{argmax}_{t \in M} B(t)$. Although the specific way to measure $B(t)$ varies across methods, it is commonly estimated based on the source trustworthiness $R(S_t)$ (i.e., level of trust assigned to the source), where $S_t = \{s : s \in S, t \in T_s\}$ is the set of sources with $t$. We compute the trustworthiness $R(s)$ and belief $B(t)$ iteratively until convergence. We used the AverageLog[53] among others (Section 1.2.1 of Supplementary Information), and the equations to update $R^i(s)$ and $B^i(t)$ for each iteration $i$ are as follows [Eq. 1–2]:

$$R^i(s) = \log |T_s| \frac{\sum_{t \in T_s} B^{i-1}(t)}{|T_s|}, \tag{1}$$

$$B^i(t) = \sum_{s \in S_t} R^i(s), \tag{2}$$

where $R^i(s)$ and $B^i(t)$ are normalized to prevent a numerical explosion by dividing with $\max_{s \in S} R^i(s)$ and $\max_{t \in T} B^i(t)$, respectively. $B^0(t)$ is set to 0.5 for all $t \in T$. Performance evaluation of alternative inconsistency resolution methods can be found in Supplementary Figs. 6–11 and Supplementary Tables 7, 8.

## Hypothesis generator

*Preprocessing.* There was not enough training data to train the hypothesis generator if we were to treat each predicate of varying temporal information distinctly. Therefore, we ultimately decided to modify the knowledge graph by removing the temporal information from the 14 predicates (e.g., 'CRA after 15 h' to CRA). This process reduced the size of the knowledge graph by 24.1% from 651,758 triples to 494,819 triples (Supplementary Data 1) and the number of predicates from 23 to 12 (Supplementary Table 4).

*Path ranking algorithm (PRA)[44,54].* The set of paired entities from the training set, linked by the CRA predicate, is first used to identify the paths used to train the model. This is done by initiating a random walk at a bounded step size starting at the subject entity. If the random walk ends up at the object entity, this path is considered successful. To reduce the size of the feature space, a path will only be considered if it links at least one object entity. Additionally, the object entity found by a path must be supported by at least a fraction, α, of the training samples. Finally, L1-regularization is used during logistic regression to reduce the feature space even more. Each path retained for the model is treated as a path feature. The value of each feature is the prior probability of reaching the object entity from the subject entity for the given sample. These path probabilities are computed recursively by assuming that every step of the path, an outgoing link to an entity, is chosen uniformly at random. After training a regularized logistic regression model to identify the parameters to these features, the final score to predict the existence of an edge in the graph is as follows [Eq. 3]:

$$score(s, o) = \sum_{P \in P_l} h_{s,P}(o) * \omega_P, \quad (3)$$

where s and o are the subject and object entities, P is one of the paths chosen by the model, $P_l$, $h_{s,P}(o)$ is the path probability, and $\omega_P$ is the weights determined using logistic regression. We set L1-regularization to 0.008, L2-regularization to 0.0001, and the fraction, α, to 0.01 based on a hyperparameter search performed on 5-fold cross-validation. More details on computing these probabilities can be found in their original work.

*Multilayer Perceptron (MLP).* The MLP, a fully connected feed-forward artificial neural network, outputs a probability of whether a given triple is true. Each entity and predicate of the knowledge graph is converted to a dense numerical vector of length 50, which is created by taking the average of the constituent word embeddings[103]. These word embeddings are randomly initialized and treated as learnable parameters for the model. A dense numerical vector of length 150, which is created by concatenating the subject, predicate, and object embeddings, is fed as an input to the network. The network contains four hidden layers, each with 60 nodes. We used ReLU[104] activation functions until the third hidden layer, followed by a Tanh activation function for the last hidden layer. Finally, the output layer uses the sigmoid activation function to produce a score between 0 and 1. We used dropout[105] after all but the last hidden layer to reduce overfitting. The model was trained to leverage the margin-based ranking loss[43] [Eq. 4]:

$$l(\omega) = \sum_{i=1}^{N} \sum_{c=1}^{C} \max\left(0, \gamma - g(T^i) + g(T^i_c)\right) + \lambda||\omega||_2^2, \quad (4)$$

where N is the number of training edges, C is the corruption size, function g() represents a complete forward pass of the network or scoring function on a given edge T, ω is the weights of the model, λ is the L2-regularization parameter, and γ is the margin that the correct edge must score higher than the corrupted edge. Based on a hyperparameter search performed on 5-fold cross-validation, we used Adam[106] optimization with a learning rate of 0.001, λ was set to 0.001, the dropout rate was set to 0.5, C was set to 100, and the margin used for training was set to 0.20.

*Stacked.* We stacked the two models PRA and MLP using AdaBoosted[107] decision stumps, in line with[57]. The training inputs to the model were three features: the scores produced by the PRA and the MLP and one binary value for the PRA to indicate whether the PRA was able to predict that certain sample. Note that the PRA cannot predict if no paths were found. We performed random search hyperparameter optimization over the validation set for each fold and found optimal parameters of 680 estimators and a learning rate of 1.65. Since our dataset is unbalanced, we also used SMOTE[108] sampling to synthetically create positive samples for a balanced set of positive and negative samples.

## Wet-lab validation

To validate whether a gene confers resistance or not, wild-type Keio strain BW25113 and its derivative single-gene knockout (KO) strains were used[109]. MIC values of the following antibiotics were measured: Amoxicillin (Sigma, Cat# A8523), Ampicillin (Roche Diagnostics, Cat# 10835269001), Apra-mycin (Alfa Aesar, Cat# AAJ6661603), Cephradine (Alfa Aesar, Cat# AAJ664960), Chloramphenicol (Calbiochem, Cat# 220551), Geneticin (Teknova, Cat# 50841719), Hygromycin B (Calbiochem, Cat# 400050100MG), Kanamycin (Acros Organics, Cat# AC611290050), Levofloxacin (Chem-Impex, Cat# 50508743), Norfloxacin (Sigma, Cat# SIAL-N9890), Novobiocin (Calbiochem, Cat# 491207), Oxycarboxine (Sigma, Cat# 36185), Paromomycin (Chem-Impex, Cat# 501602750), Rifampin (Alfa Aesar, Cat# AAJ6083603), Sisomicin (TCI, Cat#

I1049250MG), Spectinomycin (RPI, Cat# 50213656), Streptomycin (Across Organics, Cat# AC455340050), Sulfanilamide (Alfa Aesar, Cat# AAA1300122), Triclosan (Cayman Chemical Company, Cat# 501599771), Troleandomycin (Enzo Life Sciences, Cat# BML-EI249-0010), and Vancomycin (VWR Life Science, Cat# 97062-554). Since KO strains had a kanamycin resistance gene, the kanamycin resistance gene was removed from the required KO strains[110] to measure the resistance in kanamycin. Antibiotics and strains were preserved at −80 °C until used.

1 μL of required preserved strain was inoculated in 200 μL LB broth and grown overnight in an incubator shaker (BioTek Synergy HTX) at 37 °C. ~3 μL of grown culture was transferred, using a replicator, to LB agar plates containing different amounts of antibiotics, and plates were incubated overnight (~18 h) at 37 °C in an incubator. The next day, the absence and presence of colonies were monitored. The minimum concentration of antibiotic, at which no colonies were observed, was defined as MIC (Supplementary Data 8). In the case of metronidazole, colonies were observed at all concentrations. Metronidazole is a pro-drug and inactive, but in anaerobic conditions, this is converted to an active form by the bacteria[111,112]. The active form is toxic which leads to the killing of bacteria. As our experimental conditions were aerobic, metronidazole was converted to an active form, and we observed colonies at all concentrations. Subsequently, we removed metronidazole from our study.

**Reporting summary.** Further information on research design is available in the Nature Research Reporting Summary linked to this article.

## Data availability

The *E. coli* antibiotic resistance knowledge graph is available at https://github.com/IBPA/KIDS.

## Code availability

All code and instructions on how to reproduce the results can be found at https://github.com/IBPA/KIDS.

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

## Acknowledgements

We would like to thank the members of the Tagkopoulos lab and the reviewers for their suggestions, Nick Joodi and Minseung Kim for their help in the initial discussions, and Ameen Eetemadi for his comments on creating the figures. This work was supported by the USDA-NIFA AI Institute for Next Generation Food Systems (AIFS), USDA-NIFA award number 2020-67021-32855, and the NIEHS grant P42ES004699 to I.T. All code and instructions on how to reproduce the results can be found in https://github.com/IBPA/KIDS.

## Author contributions

J.Y. performed all computational analysis, and N.R. performed all wet-lab experiments. J.Y. and N.R. created the figures. J.Y., N.R., and I.T. contributed to the critical analysis and wrote the paper. I.T. conceived and supervised all aspects of the project.

## Competing interests

The authors declare no competing interests.
