## [Peer Review File · Nature Communications]

Reviewers' Comments:

Reviewer #1:

Remarks to the Author:

Summary

Youn et al. present "KIDS" an approach to discover antibiotic resistance genes via the systematic integration of data. The authors develop an approach which integrates 10 data sources into a knowledge graph, resolves inconsistencies between these data sources, and predicts new links in the graph which correspond to antibiotic resistance genes. The authors subsequently experimentally test 226 hypotheses generated by their approach to experimentally validate the performance of their model. Ultimately, the authors validate 15 novel genes leading to antibiotic resistance in E Coli.

The idea of constructing a knowledge graph specific to a particular task (i.e. antibiotic resistance gene prediction for E Coli), predicting new relationships, and ultimately testing those predictions experimentally is interesting. Similarly, the discovery of novel genes leading to antibiotic resistance in E Coli is a nice demonstration of the model's biological relevance. The authors further include extensive supplementary analyses to justify modeling decisions at each step of the work.

Nonetheless, the current version of the paper has a few major issues which require resolution.

1. First, the authors position their paper as a "systematic approach to automated knowledge discovery." This framing is much too broad for a few reasons. First, there is a wealth of prior literature that builds knowledge graphs and subsequently uses them to discover new knowledge in scientific domains. The authors are not the first to do this and the specific methodological approaches they take for the challenges they identify (i.e. inconsistency resolution, iterative prediction, link prediction) are not novel. Second, the authors only demonstrate their "highly generalizable" approach within the specific context of E Coli.

2. Second although the application domain and wet-lab experimental results are interesting, the methodological innovation of the approach is left unclear.

a. The authors point out four issues with current approaches in the introduction: unreported or unresolved conflicted information between two or more sources, a lack of negative findings, focusing on only one relation type, and inability to directly integrate results across sources. The authors' subsequent experimental results, however, do not highlight the ability of their method to solve these issues. For example, how much does the performance improve based on the authors' inconsistency resolution approach? How does that improvement compare to the prior state of the art for inconsistency resolution?

b. Similarly, the specific approaches used within each step are not methodologically novel. The inconsistency resolution and prediction approaches are applications of previously defined approaches.

3. There is critical methodological detail missing throughout the Results section of the paper. Much of this is present in the Methods section but short descriptions should be brought into the Results to make the work clearer.

Major Comments

[Figure 1] This figure is a useful overview of the paper but, as written, is too general and doesn't effectively communicate the novelty of the work. Could a bit more detail be added about the methodological approaches at each step (i.e. inconsistency resolution, hypothesis generation, validation). This comment aligns closely with the comment made in the summary. The methodological novelty of the work is unclear.

[Figure 2a] It's surprising that cellular component, biological process, and molecular function are the nodes from the Gene Ontology with the largest degree. Typically, genes are annotated to the most specific GO term possible. If the authors propagated annotations up the GO hierarchy, then every gene with a GO association will be connected to one of {biological process, molecular function, and cellular component} which seems incorrect. In a knowledge graph of this form, genes should be connected to the most specific GO terms they are associated with. Those GO

terms are then connected to more general GO terms, eventually reaching the top level {biological process, molecular function, and cellular component} nodes.

[Figure 3] Can the authors quantify exactly how much resolving these inconsistencies contributes to the overall performance of the model? Specific examples are given at Lines 97-100 but can the authors also give the systematic performance improvement (i.e. in terms of Accuracy, AUPRC, AUROC, F1)?

[Figure 3] Can the authors include additional baselines to demonstrate the strength of their approach? In particular, can the authors include a non-graph baseline. The authors should further justify their choice of MLP and PRA as their primary modeling approaches as the current description in the Supplementary (i.e. that the Knowledge Vault project uses these) is insufficient without further explanation.

[Lines 25-28] These claims should be narrowed. It is too broad to say that the approach works on all automated knowledge representation and discovery given that the results are in a highly specific context (i.e. antibiotic resistance for E Coli).

[Lines 55-58] The authors should contextualize their findings. The authors should compare their approaches to prior approaches utilizing networks/graphs for the discovery of ARGs and describe how their approach is novel. Similarly, the authors should discuss non graph based approaches.

[Figure 5a; Lines 111-112] Far more hypotheses were tested in the low probability range [0.0, 0.2] than in the high probability range (0.8, 1.0) which seems counterintuitive (i.e. in an actual setting, wouldn't the top predictions be the most tested)? The performance score here may be inflated since, as the authors point out in Figure 2E, there are far more negative predicates than positive predicates. As a result, the model may find it easier to predict negatives than positives.

[Figure 5a] The reported R2 value seems to be generated on only 5 points which is not robust.

[Lines 150-151] The authors position their work as a broad, flexible framework for knowledge integration and discovery. However, they only demonstrate the utility of the framework on a specific, narrow problem (antibiotic resistance genes for E Coli). Given that other methods have, in many other contexts, developed frameworks to integrate and discovery knowledge via knowledge graphs, this doesn't seem like an appropriate claim.

[Lines 151-155] The authors should position their method, not as a solution to the general problem of antibiotic resistance genes, but as a more specific improvement over prior methods in this space (i.e. those that use graphs and others).

Additional comments, mostly to improve clarity of writing

[Lines 92-94] A sentence of two about about the "ground truth wet-lab validation" and how it was generated should be included.

[Lines 21-22] This description of the results in the Abstract should be revised to improve clarity.

[Lines 76-81] More detail should be included about the types of data that were included and the types of resulting relations.

[Lines 88-100] The authors should include more methodological details about how inconsistency resolution was handled in the Results not just the methods.

[Lines 101-110] What is the task here? The authors should more clearly state the task, not just in the caption, but also in the main text. Use of the phrase "link prediction" would help as well as well as a description of the operating point selected for each model, how that operating point was selected, and what that ultimately means for the generation of the Precision, Recall, AUPRC, and F1 scores.

[Lines 111-125] Methodological detail about the wet-lab validation should be included. Describing the experiments is critical as it determines the evaluation of the model.

[Lines 160-168] This description of which models were selected and why they were selected should be in the Results section.

[Lines 209] Can the authors describe exactly how these rules were generated?

[Lines 212-213] Can the authors describe exactly how these rules were generated?

[Lines 219-220] Why is the assumption that t belongs to one unique M_t valid?

[Lines 234] A reference to the Supplementary Results evaluating the performance of the inconsistency resolution methods would be helpful.

[Lines 203] How exactly was this "falsification" done? How were incorrect triples generated?

[Lines 259-261, 278-280] How were these parameters chosen?

[Lines 390-402] The interpretable nature of PRA is nice. The authors may consider highlighting this earlier.

[SF1] Given the block-like structure of this figure, the authors could consider evaluating the model's performance on antibiotics with very few positive/negative labels. While not necessary to do, this could be an interesting "few shot" learning result.

[Figure 5b] Is there a legend or part of the caption missing? It is hard to understand this figure. What does each bar represent?

[Figure 5c, d] It is nice to see the performance improvement across cycles. This should be highlighted as a larger contribution of your approach compared to prior work.

Reviewer #2:

Remarks to the Author:

This paper develops a pipeline platform that integrates different data sources, builds a temporary knowledge graph, solves inconsistency to refine the knowledge graph, and uses the inconsistency-free knowledge graph to generate hypotheses that are further validated via wet-lab test. The main contribution is the construction of an inconsistency-free knowledge graph that supports multiple triple types and multiple data sources and performs knowledge graph link prediction for hypothesis generation.

Knowledge graph embedding model

This paper applies models including PRA, MLP, and their stacking, which are a bit outdated. More representative models are missing from experiments such as bilinear models, factorization-based models, and more state-of-the-art neural models. Leaving them for further study is not good enough as a submission to a prestigious journal. As in 168-169, the cited papers are too general in the context of the knowledge graph. There are many specific works linked to knowledge graph embedding and attention modules used in the field of knowledge graph as reviewed in recent surveys (Knowledge graph embedding: A survey of approaches and applications, IEEE TKDE 2017 & A survey on knowledge graphs: Representation, acquisition and applications, IEEE TNNLS 2021). The experimental study should also test recent advances.

Novelty

What is the distinguishable novelty? There are several domain-specific applications of knowledge graphs, for example, proteomics and cancer (KnowLife: a versatile approach for constructing a large knowledge graph for biomedical sciences, BMC Bioinformatics 2015; Clinical Knowledge Graph Integrates Proteomics Data into Clinical Decision-Making, bioRxiv 2020 & Metastatic Site Prediction in Breast Cancer using Omics Knowledge Graph and Pattern Mining with Kirchhoff's Law Traversal, bioRxiv 2020). Except for the application in a different domain, what is the difference? This paper also resolves the inconsistency in multiple sources. What is the novelty of inconsistency resolver compared with existing methods on inconsistency resolution?

Generalizability

The claim in lines 181-183 is subjective to some extent. Or at least, no strong evidence is observed in this paper to support the generalizability of the proposed framework. If the generalizability is inherited from the knowledge graph-based model, it shouldn't be the contribution of the proposed framework for a specific application in life science. Besides, a domain-specific knowledge graph is limited to a specific domain. If the claim is for the procedures of knowledge construction and inference, this seems to be a common practice. Perhaps this claim needs further clarification.

Data integration

Ten different knowledge sources are selected. How many sources are reviewed as the selection

pool? What is the selection criterion?

Knowledge inference rules

How knowledge inference rules (sup. Data 6) are obtained? How do make sure (as much as possible) that they are complete?

Temporal information

In preprocessing, temporal information is removed due to data scarcity. Will there be any side effects?

Challenges

As written in lines 43-46, what does "lack of negative findings" mean? How is it addressed in this paper?

Overall, this is a novel domain application of knowledge graph, and this paper constructs a knowledge graph and conducts a good experimental study with their knowledge graph-based platform. The methodological novelty of knowledge graph construction and representation learning needs to be enhanced or clarified and the empirical comparison with recent advances in knowledge graph embedding should also be included.

Reviewer #3:

Remarks to the Author:

My concerns

1) This method uses knowledge available on genes to classify them whether they are antibiotic resistance genes. This method is very good to deal with well-documented organisms such as E.col and its genes. Dealing with new organisms and different genes of unknown function, for example, the method will fail to detect whether they confer resistance or not.

2) The experimental framework is not fully covered. Authors need to do a whole-genome sequencing of the organisms they test to ensure that the phenotypic resistance observed is not an artifact of mutations on other regions of the genome. Evolutionary elements such as mutations and indels could also lead to the gain or loss of antibiotic resistance phenotypes.

3) Authors did not show how similar these genes are at the sequence level to already known antibiotic resistance genes.

4) Authors did not show how much dissemination these genes have across microbial communities. Such information could help us decided whether these genes are clinically relevant antibiotic resistance genes or not.

5) Authors should also test these genes on different organisms such as Bacillus subtilis

If the authors could address my concerns, I will be happy to have another very in-deep review of this article.

Reviewer #1

Summary

Youn et al. present “KIDS” an approach to discover antibiotic resistance genes via the systematic integration of data. The authors develop an approach which integrates 10 data sources into a knowledge graph, resolves inconsistencies between these data sources, and predicts new links in the graph which correspond to antibiotic resistance genes. The authors subsequently experimentally test 226 hypotheses generated by their approach to experimentally validate the performance of their model. Ultimately, the authors validate 15 novel genes leading to antibiotic resistance in E Coli.

The idea of constructing a knowledge graph specific to a particular task (i.e. antibiotic resistance gene prediction for E Coli), predicting new relationships, and ultimately testing those predictions experimentally is interesting. Similarly, the discovery of novel genes leading to antibiotic resistance in E Coli is a nice demonstration of the model’s biological relevance. The authors further include extensive supplementary analyses to justify modeling decisions at each step of the work.

Nonetheless, the current version of the paper has a few major issues which require resolution.

1. First, the authors position their paper as a “systematic approach to automated knowledge discovery.” This framing is much too broad for a few reasons. First, there is a wealth of prior literature that builds knowledge graphs and subsequently uses them to discover new knowledge in scientific domains. The authors are not the first to do this and the specific methodological approaches they take for the challenges they identify (i.e. inconsistency resolution, iterative prediction, link prediction) are not novel. Second, the authors only demonstrate their “highly generalizable” approach within the specific context of E Coli.

Response: We agree with the reviewer that there exists prior work that uses knowledge graphs to discover new knowledge. We extended the references provided in this topic by adding 4 more¹⁻⁴ and added in the discussion of this topic (lines 61-63) to further clarify this point. Although we do not claim novelty on these methods, this work led to the construction of the largest inconsistency-free KG in the biological domain for a single bacterium and a novel application of the knowledge graph-based approach to the discovery of new ARGs. We have revised the text in the introduction (lines 70-72) and discussion (lines 201-209) to this end. Finally, it is clear that this methodology can be applied to other bacteria or other ontological groups beyond ARGs, but since we have not demonstrated it in this work, we have removed the statement in the revised manuscript, and instead listed this as future work for evaluating how broad the method can be applied (lines 257-260).

2. Second although the application domain and wet-lab experimental results are interesting, the methodological innovation of the approach is left unclear.

a. The authors point out four issues with current approaches in the introduction: unreported or unresolved conflicted information between two or more sources, a lack of negative findings, focusing on only one relation type, and inability to directly integrate results across sources. The authors' subsequent experimental results, however, do not highlight the ability of their method to solve these issues. For example, how much does the performance improvements based on the authors' inconsistency resolution approach? How does that improvement compare to the prior state-of-the-art for inconsistency resolution?

Response: We now highlight the performance gain for solving each of these issues. In the case of *conflicting information* between two or more sources, we identify and resolve 236 sets of inconsistencies despite their identical experimental settings^{5,6} in lines 101-105. While on the summary statistics the improvement from resolved inconsistencies is small, as expected due to their small size when compared to the knowledge graph, we found 2 new antibiotic-resistant relationships (*surA*, CRA, Vancomycin) and (*asmA*, CRA, Vancomycin) only after reinstating the resolved inconsistencies into the knowledge graph (lines 116-119), something that demonstrates the importance of inconsistency resolution and coherence in our knowledge. For the lack of negative findings, our knowledge graph is the first to include both the positive findings (14 triple types, 31,216 triples) and the negative findings (9 triple types, 620,542 triples) to the best of our knowledge. Although the majority of the hypothesis generation models we tested did not use these negatives and instead generated them either through closed-world-assumption (PRA) or corruption through random sampling (MLP, TransE, TransD, and TuckER), our best model (Stacked) did utilize these negatives. We believe there is still a potential to take advantage of these negative findings in other machine learning models. To address the focus on only one relation type, our knowledge graph contains 23 relation types (Supplementary Table 2) as opposed to a single relation type from other sources (Supplementary Section 1.1.1). Regarding the inability to directly integrate results across sources due to incompatible metadata, this is still a problem for this and any other framework, as it is related to data incompatibility during their generation and reporting. We now address these points in the discussion section lines 213-234.

b. Similarly, the specific approaches used within each step are not methodologically novel. The inconsistency resolution and prediction approaches are applications of previously defined approaches.

Response: We want to emphasize that we do not claim the novelty of the specific methods used in both the inconsistency resolution and hypothesis generation modules. For the inconsistency resolution methods, we tested 6 different versions (Supplementary Section 1.2.1) and chose the AverageLog method based on evaluation on a synthetic dataset (Supplementary Section 1.2.2). For the hypothesis generation methods, in addition to the PRA, MLP, and Stacked model, we now test three additional methods (TransE, TransD, TuckER). This work, however, led to the construction of the largest inconsistency-free KG in the biological domain for a single bacterium and a novel application of the knowledge graph-based approach to the discovery of new ARGs. We have revised the text in the introduction (lines 70-72) and discussion (lines 201-209) to this end.

3. There is critical methodological detail missing throughout the Results section of the paper. Much of this is present in the Methods section but short descriptions should be brought into the Results to make the work clearer.

Response: As suggested by the reviewer in the minor comments, we added methodological details throughout the results section of the manuscript. In lines 86-91, we now discuss the types of input raw data as well as the resulting triple types. In lines 105-108, we describe what method of inconsistency resolver is used, as well as a brief description of how the resolver is trained. In lines 111-114, we added how the ground truth wet-lab validation was performed. Finally, in lines 120-135, we added what the task of the hypothesis generator is and the description of each hypothesis generator method.

Major Comments

[Figure 1] This figure is a useful overview of the paper but, as written, is too general and doesn't effectively communicate the novelty of the work. Could a bit more detail be added about the methodological approaches at each step (i.e. inconsistency resolution, hypothesis generation, validation). This comment aligns closely with the comment made in the summary. The methodological novelty of the work is unclear.

Response: We have now revised Figure 1 to include more details of each step of KIDS. This is an overarching figure, with a more detailed illustration of each step in Figures 3 (inconsistency resolution) and 4 (hypothesis generation).

[Figure 2a] It's surprising that cellular component, biological process, and molecular function are the nodes from the Gene Ontology with the largest degree. Typically, genes are annotated to the most specific GO term possible. If the authors propagated annotations up the GO hierarchy, then every gene with a GO association will be connected to one of {biological process, molecular function, and cellular component} which seems incorrect. In a knowledge graph of this form, genes should be connected to the most specific GO terms they are associated with. Those GO terms are then connected to more general GO terms, eventually reaching the top level {biological process, molecular function, and cellular component} nodes.

Response: We confirm what the reviewer suggests, that genes are connected to the most specific GO terms they are associated with. We would like to clarify that 'cellular component', 'biological process', and 'molecular function' are not the nodes in our knowledge graph but just text labels that denote the node type of each axis in the hive plot of Figure 2A. The actual nodes with the largest degree from each node type are indeed specific GO terms '*cytosol*', '*cellular response to DNA damage stimulus*', and '*response to antibiotic*', as it is shown in Figure 2A. We have now included this point in the figure caption for disambiguation.

[Figure 3] Can the authors quantify exactly how much resolving these inconsistencies contributes to the overall performance of the model? Specific examples are given at Lines 97-100 but can the authors also give the systematic performance improvement (i.e. in terms of Accuracy, AUPRC, AUROC, F1)?

Response: Please see the response to question 2a. Comparing the performance of the hypothesis generator before/after reinstating the resolved inconsistencies, did not show any statistically significant performance improvement on a broader spectrum in terms of the mentioned metrics other than the specific examples presented in lines 116-119. This is most likely because only 7 out of the 236 inconsistencies (3.0%) we experimentally resolved and further validated in the wet lab were positive triples (Supplementary Data 2), and therefore reinstating them back to the knowledge graph where 1,606 positive CRA triples exist (Supplementary Data 1) does not affect the knowledge graph much (increase from 1,606 to 1,613, a 0.44% increase). Moreover, as our models do not fully utilize the negatives in the training phase and instead use closed-world assumption or corruption through random sampling to generate synthetic negatives, we did not observe any significant performance improvement. However, the resolution of inconsistencies led to novel insights as mentioned in response 2a and the discussion section of the manuscript (lines 218-228).

[Figure 3] Can the authors include additional baselines to demonstrate the strength of their approach? In particular, can the authors include a non-graph baseline. The authors should further justify their choice of MLP and PRA as their primary modeling approaches as the current description in the Supplementary (i.e. that the Knowledge Vault project uses these) is insufficient without further explanation.

Response: In this work, we only focused on and tested the graph-based machine learning methods instead of non-graph methods like culture-based methods and whole-genome sequencing analysis

methods (lines 57-60). We now have tested three more recent knowledge graph completion methods ⁷, TransD⁸, and TuckER⁹. Both TransE and TransD had worse F1-score of 21.7% and 23.7%, respectively, compared to the Stacked model with 30.1%. TuckER, which is a factorization-based model that was introduced after the project was conceived, did have a slightly higher F1-score compared to the Stacked model (30.8% vs. 30.1%), but the results were statistically insignificant (p -value: 0.65). We now include all of these results in Figure 4, the results section (lines 123-133 and 662-665), Supplementary Information sections 1.3.5 and 1.3.6, and Supplementary Table 5.

[Lines 25-28] These claims should be narrowed. It is too broad to see that the approach works on all automated knowledge representation and discovery given that the results are in a highly specific context (i.e. antibiotic resistance for E Coli).

Response: We have removed the generalizability statement in the revised manuscript, and instead listed this as future work for evaluating how broad the method can be applied (lines 257-258).

[Lines 55-58] The authors should contextualize their findings. The authors should compare their approaches to prior approaches utilizing networks/graphs for the discovery of ARGs and describe how their approach is novel. Similarly, the authors should discuss non graph based approaches.

Response: We now mention the previous research and further clarify our novelty in the introduction lines 61-66. We also further discuss non-graph-based approaches and compare them with KIDS in the discussion section (lines 201-209). To the best of our knowledge, our approach is the first to perform knowledge graph completion (KGC) on the knowledge graphs to discover novel ARGs. In the biological domain, however, we are aware of several applications of KGC for the discovery of new biological knowledge^{1-4,10}, which we now include in the revised manuscript (lines 61-64).

[Figure 5a; Lines 111-112] Far more hypotheses were tested in the low probability range [0.0, 0.2] than in the high probability range (0.8, 1.0) which seems counterintuitive (i.e. in an actual setting, wouldn't the top predictions be the most tested)? The performance score here may be inflated since, as the authors point out in Figure 2E, there are far more negative predicates than positive predicates. As a result, the model may find it easier to predict negatives than positives.

Response: It is correct that one would test the top predictions more in the real-world scenario, however, here we wanted to have a more balanced representation of the probability bins. For our case, we first decided to bin the first iteration hypotheses into 5 groups. We then moved on to cover at least 70% of the hypotheses with a probability greater than 0.2 (we ultimately tested 105 out of 149 first iteration hypotheses ≥ 0.2 , a 70.5% coverage). We decided to test more hypotheses in the lowest probability range [0.0, 0.2] to better statistically reflect its larger population size (107,929 hypotheses). As for the concern regarding performance scores being inflated due to class imbalance, we do upsample the minority class using SMOTE¹¹ to make sure that the model does not get biased towards the negative class (Methods, lines 370-371).

[Figure 5a] The reported R2 value seems to be generated on only 5 points which is not robust.

Response: We report the $R^2=0.94$ by first binning the probability of the 316 hypotheses from both iterations (226 and 90 from first and second iterations, respectively) into 5 bins. We initially tried a larger number of bins (10 to be specific) but found some bins with not many samples (e.g., (0.8, 0.9) only has 5 samples) and therefore decided to use 5 bins. We do, however, agree that we should show all analyses performed on the dataset, and we now include the same analysis but with 10 bins in Supplementary Section 1.3.9 (lines 562-564) and Supplementary Figure 16. The results show that the probability of positive antibiotic-specific findings still highly correlates with experimentally validated findings ($R^2=0.92$).

[Lines 150-151] The authors position their work as a broad, flexible framework for knowledge integration and discovery. However, they only demonstrate the utility of the framework on a specific, narrow problem (antibiotic resistance genes for E Coli). Given that other methods have, in many other contexts, developed frameworks to integrate and discovery knowledge via knowledge graphs, this doesn't seem like an appropriate claim.

Response: We agree there exist previous works in the broader domain of biomedical science that constructs knowledge graphs and performs knowledge graph completion for the discovery of new knowledge^{4,10,12} similar to what we do in this manuscript. We, therefore, removed any statements about the generalizability of our proposed work and modified the introduction (lines 61-64) to discuss these previous works.

[Lines 151-155] The authors should position their method, not as a solution to the general problem of antibiotic resistance genes, but as a more specific improvement over prior methods in this space (i.e. those that use graphs and others).

Response: KIDS is an addition to the methods addressing the discovery of ARGs. Previous non-graph-based approaches (Introduction, lines 57-60) required gene sequencing information as input to the computational models. For example, best hit-based methods identify potential ARGs in the genomic and metagenomic sequences based on their similarity to the known ARGs in the existing databases. These approaches, however, will fail to identify ARGs if the sequence similarity is low or the sequences of known ARGs are absent in the reference database. Compared to these methods that rely on sequencing information, the power of the KIDS framework stems from guilt-by-association and pattern discovery within the knowledge graph. To show this, we downloaded the nucleotide sequence of the 4,577 ARGs from CARD (version 3.1.4) and performed nucleotide BLAST (blast.ncbi.nlm.nih.gov/Blast.cgi) with our 6 ARGs (*ftsP*, *hdfR*, *Irp*, *proV*, *qorB*, and *rbsK*). However, we did not identify any statistically significant homologs (E-value < 0.05), while the best hit was for *OXA-541* of *Pseudomonas putida* for *Irp* (91.7% sequence similarity, E-value = 0.12). Please note that out of the 129 genes from the lowest probability range [0.0, 0.2] that we have validated to have no predicted ARG activity, we found 9 genes that have significant E-value (< 0.05) with >68.6% sequence similarity, arguing that just looking at homology is not sufficient for discovering ARGs. We updated the manuscript to reflect these findings (Results, lines 180-183; Supplementary Information, Section 1.3.12; Supplementary Data 10; Discussion, lines 201-209).

Additional comments, mostly to improve clarity of writing

[Lines 92-94] A sentence of two about the “ground truth wet-lab validation” and how it was generated should be included.

Response: We have added the following texts in lines 111-114: *‘... which was performed by measuring and comparing the minimum inhibitory concentrations (MICs) of the single-gene knock-out strain and the wild-type strain on the LB agar plate (Supplementary Information Section 1.2.4, Supplementary Data 8).’*

[Lines 21-22] This description of the results in the Abstract should be revised to improve clarity.

Response: We have revised the description of the results in the abstract.

[Lines 76-81] More detail should be included about the types of data that were included and the types of resulting relations.

Response: We modified the manuscript to add more detail regarding the types of data included in the knowledge graph (lines 86-91) as well as a reference to the relevant section (Supplementary Information, Section 1.1.1). We also added more detail regarding the types of resulting relations at lines 93-97.

[Lines 88-100] The authors should include more methodological details about how inconsistency resolution was handled in the Results not just the methods.

Response: We modified the corresponding results section (lines 105-108) to make the steps of inconsistency resolution more clear as well as including more details of the methodology by specifying what resolution method was used, how it was trained, and by putting a reference to Figure 3B earlier in the paragraph.

[Lines 101-110] What is the task here? The authors should more clearly state the task, not just in the caption, but also in the main text. Use of the phrase “link prediction” would help as well as a description of the operating point selected for each model, how that operating point was selected, and what that ultimately means for the generation of the Precision, Recall, AUPRC, and F1 scores.

Response: The task of the hypothesis generator is indeed to perform link prediction on the missing CRA links between all pairwise combinations of *E. coli* genes and antibiotics in the knowledge graph. We now clearly state what the task of the HG is in lines 120-122 using the more commonly used term ‘link prediction’ as suggested by the reviewer. As for the choice of the operating point selected for each model, we optimized all models using the F1 score. In other words, the PR curve, AUCPR, and all statistics (Figure 4C and Supplementary Table 5) were generated using the models optimized using the F1 score. We clarify this point in the manuscript in lines 135-136.

[Lines 111-125] Methodological detail about the wet-lab validation should be included. Describing the experiments is critical as it determines the evaluation of the model.

Response: The wet-lab validation of the generated hypotheses was conducted similar to that in the inconsistency resolver (lines 111-114) by measuring and comparing the minimum inhibitory concentrations (MICs) of the single-gene knock-out strain and the wild-type strain on the LB agar plate. We updated the manuscript to clarify this point and added a reference to the relevant section in the Supplementary Information.

[Lines 160-168] This description of which models were selected and why they were selected should be in the Results section.

Response: We now discuss methodological details of the hypothesis generation methods PRA, MLP, Stacked, TransE, and TransD in the results section (lines 123-133). Moreover, we added a new paragraph (lines 235-243) in the discussion to review the state-of-the-art link prediction methods as well as plans to utilize them in future work.

[Lines 209] Can the authors describe exactly how these rules were generated?

Response: We came up with 15 sets of knowledge inference rules upon visual inspection of the 23 triple types as stated in the revised manuscript (lines 282-286). There are automatic knowledge graph completion methods^{13,14} that can potentially do this automatically, but we leave it for future work as their precision is not at human-level yet nor have been tested in the biomedical domain, which can ultimately cause negative effects. We discuss this point in the discussion section (lines 244-248).

[Lines 212-213] Can the authors describe exactly how these rules were generated?

Response: Similar to the knowledge inference rules, we clarified in the manuscript that we manually defined 9 sets of inconsistency detection rules after inspection of the knowledge graph. We also added an example of a sample inconsistency in lines 292-294.

[Lines 219-220] Why is the assumption that t belongs to one unique M_t valid?

Response: The word choice of 'assume' was misleading in this case. We updated the corresponding part as follows: *'In an inconsistency-free setting, a triple t belongs to one unique M_t . In other words, $|M_t| = 1$ means there exist no conflicts in M_t .'*

[Lines 234] A reference to the Supplementary Results evaluating the performance of the inconsistency resolution methods would be helpful.

Response: We added a sentence to refer the readers to Supplementary Figures 6-11 and Supplementary Tables 7 and 8.

[Lines 203] How exactly was this “falsification” done? How were incorrect triples generated?

Response: In hiTRN¹⁵, there exist four predicate types (activates, represses, no activates, no represses). We generated the incorrect triples by replacing the predicate of the triple with its negative counterpart (e.g., ‘no activates’ -> ‘activates’ and ‘represses’ -> ‘no represses’). We modified the supplementary information to reflect this clarification (Supplementary Information, lines 224-228).

[Lines 259-261, 278-280] How were these parameters chosen?

Response: We performed a hyperparameter search using 5-fold cross-validation. We updated the manuscript to reflect this information.

[Lines 390-402] The interpretable nature of PRA is nice. The authors may consider highlighting this earlier.

Response: We added more methodological descriptions about the PRA in the results section (lines 128-130) emphasizing its interpretability as well as providing reference to Supplementary Table 9 that shows the actual path features used by PRA to generate hypotheses.

[SF1] Given the block-like structure of this figure, the authors could consider evaluating the model’s performance on antibiotics with very few positive/negative labels. While not necessary to do, this could be an interesting “few shot” learning result.

Response: We agree that evaluating the model’s performance on those antibiotics with few known labels in a few-shot learning manner would give us insight into how the KIDS framework generalizes to other bacterial species with limited training data. Since we modified the generalization claim, we added a section in the discussion (lines 258-260) to address this point. Thank you for this suggestion.

[Figure 5b] Is there a legend or part of the caption missing? It is hard to understand this figure. What does each bar represent?

Response: From the second cycle of hypothesis generation, we validated 29 out of 90 hypotheses to be positive (CRA). The box plot on the right shows the probability distribution of these 29 positively validated hypotheses from the second cycle (i.e., dark blue bars in Figure 5A), whereas the box plot on the left shows the probability distribution of those identical 29 hypotheses from the ‘first’ cycle of hypothesis generation. This plot shows how updating the knowledge graph through multi-cycle hypothesis generation enables the discovery of new hypotheses that were hidden (due to a low

probability) in the initial iteration. We updated the figure legend to reflect this updated information in lines 673-680.

[Figure 5c, d] It is nice to see the performance improvement across cycles. This should be highlighted as a larger contribution of your approach compared to prior work.

Response: We now stress the ‘multi-cycle hypothesis generation’ in lines 25 of the abstract and lines 155-156 of the results section. Moreover, we clarify that 1 out of the 6 novel ARG discoveries came from the first iteration of hypothesis generation, whereas the remaining 5 came from the second iteration in line 25 of the abstract and lines 159-160 of the results section.

We would like to thank the first reviewer for insightful feedback!

Reviewer #2

This paper develops a pipeline platform that integrates different data sources, builds a temporary knowledge graph, solves inconsistency to refine the knowledge graph, and uses the inconsistency-free knowledge graph to generate hypotheses that are further validated via wet-lab test. The main contribution is the construction of an inconsistency-free knowledge graph that supports multiple triple types and multiple data sources and performs knowledge graph link prediction for hypothesis generation.

Knowledge graph embedding model

This paper applies models including PRA, MLP, and their stacking, which are a bit outdated. More representative models are missing from experiments such as bilinear models, factorization-based models, and more state-of-the-art neural models. Leaving them for further study is not good enough as a submission to a prestigious journal. As in 168-169, the cited papers are too general in the context of the knowledge graph. There are many specific works linked to knowledge graph embedding and attention modules used in the field of knowledge graph as reviewed in recent surveys (Knowledge graph embedding: A survey of approaches and applications, IEEE TDKE 2017 & A survey on knowledge graphs: Representation, acquisition and applications, IEEE TNNLS 2021). The experimental study should also test recent advances.

Response: We have now expanded to additional state-of-the-art models. As suggested, we tested 5 additional methods TransE⁷, TransD⁸, Simple¹⁶, RotatE¹⁷, and TuckER⁹. Experimental results show that TransE and TransD have F1 scores of 21.7% and 23.7%, respectively, which is lower than the MLP (28.9%) or the Stacked (30.1%). Even after a grid-search of hyperparameters, we were not able to obtain a proper set of hyperparameters for Simple and RotatE that performs better than our worst performing model PRA (14.6%). For the factorization-based model TuckER, we obtained an F1 score of 30.8% which is 0.7% higher than the best model Stacked (30.1%), but the results were statistically insignificant (p -value: 0.65). We updated Figure 4, Results section (lines 123-135), Supplementary Information Sections 1.3.5 and 1.3.6, and Supplementary Table 5 to include these newly tested hypothesis generation methods. To address the reviewer's comment about cited papers being too general, we now discuss the link prediction task in detail (lines 120-124) and specific representative models both in the introduction (lines 63-66) and discussion (lines 235-243).

Novelty

What is the distinguishable novelty? There are several domain-specific applications of knowledge graphs, for example, proteomics and cancer (KnowLife: a versatile approach for constructing a large knowledge graph for biomedical sciences, BMC Bioinformatics 2015; Clinical Knowledge Graph Integrates Proteomics Data into Clinical Decision-Making, bioRxiv 2020 & Metastatic Site Prediction in Breast Cancer using Omics Knowledge Graph and Pattern Mining with Kirchhoff's Law Traversal, bioRxiv 2020). Except for the application in a different domain, what is the difference? This paper also

resolves the inconsistency in multiple sources. What is the novelty of inconsistency resolver compared with existing methods on inconsistency resolution?

Response: We agree with the reviewer that there exists prior work that uses knowledge graphs to discover new knowledge. We extended the references provided in this topic by adding 4 more¹⁻⁴ (Sang et al., 2018; Segler and Waller, 2017; Hassani-Pak and Rawlings, 2017; Santoa et al., 2020) and added in the discussion of this topic (lines 61-66) to further clarify this point. Although we do not claim novelty on the inconsistency resolution methods nor the hypothesis generation methods, this work is the first to this application and has led to the construction of the largest inconsistency-free KG in the biological domain for a single bacterium and a novel application of the knowledge graph-based approach to the discovery of new ARGs. We have revised the text in the introduction (lines 67-72) and discussion (lines 201-209) to this end. We also now emphasize the performance improvement coming from the ‘multi-cycle hypothesis generation’ (lines 23-25 of the abstract and lines 155-156 of the results section). Although we do not claim the methodological novelty of the inconsistency resolution methods, we do show that the resolved inconsistencies led to the discovery of two antibiotic-resistant relationships (*surA*, CRA, Vancomycin) and (*asmA*, CRA, Vancomycin), that would otherwise be left undiscovered (lines 116-119).

Generalizability

The claim in lines 181-183 is subjective to some extent. Or at least, no strong evidence is observed in this paper to support the generalizability of the proposed framework. If the generalizability is inherited from the knowledge graph-based model, it shouldn't be the contribution of the proposed framework for a specific application in life science. Besides, a domain-specific knowledge graph is limited to a specific domain. If the claim is for the procedures of knowledge construction and inference, this seems to be a common practice. Perhaps this claim needs further clarification.

Response: It is clear that this methodology can be applied to other bacteria or other ontological groups beyond ARGs, but since we have not demonstrated it in this work, we have removed the statement in the revised manuscript. Instead, we listed this as future work for evaluating how broad the method can be applied (lines 257-258).

Data integration

Ten different knowledge sources are selected. How many sources are reviewed as the selection pool? What is the selection criterion?

Response: We performed a literature search regarding *E. coli* antibiotic resistance profiling, and the sources we provided in this manuscript were the best we could find. We did have 4 additional sources¹⁸⁻²¹ but ultimately decided to not include them in the knowledge graph as the experimental condition was very different from the existing ones. We included sources like GO and hiTRN in our knowledge graph to enrich the knowledge graph with more diverse predicate types and thus allow models like PRA to perform properly.

Knowledge inference rules

How knowledge inference rules (sup. Data 6) are obtained? How do make sure (as much as possible) that they are complete?

Response: Knowledge inference rules (Supplementary Data 6) were generated by closely inspecting the existing triple types and considering all possible rule combinations. We now clarify this point in the manuscript (lines 282-286). One possible automated approach for inference rule generation is to use automatic knowledge graph construction methods like COMET¹⁴, with a potential pitfall of creating noise in the data (lines 244-248), therefore negatively affecting the performance of the hypothesis generators. We now address this point in the Supplementary Information Section 1.1.2 (lines 153-164).

Temporal information

In preprocessing, temporal information is removed due to data scarcity. Will there be any side effects?

Response: As the reviewer pointed out, the removal of temporal information was an inevitable decision due to the lack of training data if we were to treat each predicate with temporal information unique (e.g., 'CRA after 7 days' only has 59 triples, Supplementary Table 2). The side effect of removing the temporal information is the creation of inconsistencies. For example, although the two triples (*cydX*, CRA after 15 hours, Vancomycin) and (*cydX*, -CRA after 18 hours, Vancomycin) supported by Nichols et al.²² and Tamae et al.⁵, respectively, are not inconsistencies in their original form, they become inconsistencies after removing the temporal information. This phenomenon is discussed in detail in the Supplementary Information Section 1.2.3. We also added a new section in the Supplementary Information (lines 393-405) discussing this side effect in detail.

Challenges

As written in lines 43-46, what does "lack of negative findings" mean? How is it addressed in this paper?

Response: The majority of the sources used to curate our knowledge graph only contain positive data such as positive CRA but not negative CRA, which is due to a common practice in the field of microbiology of only reporting the positive findings. As these 'negative findings' are important when training any machine learning methods, we address this challenge in our work by extracting the negative findings from the raw data (Supplementary Information Section 1.1.1). Thus, our knowledge graph contains not only the positive findings (9 triple types) but also the negative findings (14 triple types). We modified the manuscript in the introduction (line 47 and lines 70-72) and results (lines 93-95) to clarify these points. Although the majority of the hypothesis generation models we tested did not use these negatives and instead generated them either through closed-world-assumption (PRA) or corruption through random sampling (MLP, TransE, TransD, and TuckER), our best model (Stacked) did utilize these negatives (lines 364-371). We believe there is still a potential to take advantage of these negative findings in other machine learning models (lines 221-228).

Overall, this is a novel domain application of knowledge graph, and this paper constructs a knowledge graph and conducts a good experimental study with their knowledge graph-based platform. The methodological novelty of knowledge graph construction and representation learning needs to be enhanced or clarified and the empirical comparison with recent advances in knowledge graph embedding should also be included.

Response: We now compare to recent advances in KG embeddings methods. We also tested 5 additional methods TransE⁷, TransD⁸, Simple¹⁶, RotatE¹⁷, and TuckER⁹. Experimental results show that TransE and TransD have F1 scores of 21.7% and 23.7%, respectively, which is lower than the MLP (28.9%) or the Stacked (30.1%). Even after a grid-search of hyperparameters, we were not able to obtain a proper set of hyperparameters for Simple and RotatE that performs better than our worst performing model PRA (14.6%). For the factorization-based model TuckER, we obtained an F1 score of 30.8% which is 0.7% higher than the best model Stacked (30.1%), but the results were statistically insignificant (p -value: 0.65). We updated Figure 4, Results section (lines 123-140), Supplementary Information Sections 1.3.5 and 1.3.6, and Supplementary Table 5 to include these newly tested hypothesis generation methods. We modified the discussion section (lines 235-243) to cover the state-of-the-art link prediction methods as well as plans to test them in future work.

We would like to thank the second reviewer for insightful feedback!

Reviewer #3

1) This method uses knowledge available on genes to classify them whether they are antibiotic resistance genes. This method is very good to deal with well-documented organisms such as E.col and its genes. Dealing with new organisms and different genes of unknown function, for example, the method will fail to detect whether they confer resistance or not.

Response: The reviewer is right. The method will be limited in its ability to generalize in organisms with less information. However, in that case, gene homology and transfer learning from model organisms such as *E. coli* will help in the KG augmentation and completion task, something that we are not addressing in this paper. It is clear that this methodology can be applied to other bacteria or other ontological groups beyond ARGs as the generalizability roots from the use of knowledge graph, but since we haven't demonstrated it in this work, we have removed the statement in the revised manuscript. Instead, we listed this as future work for evaluating how broad the method can be applied (lines 257-258). With regards to the reviewer's comment regarding the application of the framework on organisms that are not well-studied like *E. coli*, we include a discussion regarding the future work (lines 258-260), to see how the framework would generalize to other organisms with limited training data.

2) The experimental framework is not fully covered. Authors need to do a whole-genome sequencing of the organisms they test to ensure that the phenotypic resistance observed is not an artifact of mutations on other regions of the genome. Evolutionary elements such as mutations and indels could also lead to the gain or loss of antibiotic resistance phenotypes.

Response: We measured the MICs on LB agar plates containing different concentrations of antibiotics, where growths of wild-type (WT) and single gene-knockout (KO) strains were documented after overnight growth (16-18h) at 37°C in three biological replicates. Minimum antibiotic concentration inhibiting the growth of all three biological replicates was picked up as MIC. We agree that resistant mutations can emerge over a short period, however in our experimental setting, as described above, chances of the selection of a resistant phenotype emerging from the spontaneous mutation are small. Moreover, our method is one of the established microbiological methods for the determination of MIC and has been used at least by two of our largest knowledge sources^{5,6}, since while WGS would be the golden standard for ensuring lack of facultative gain-of-function mutations, it is impractical and with limited value over just MIC determination.

3) Authors did not show how similar these genes are at the sequence level to already known antibiotic resistance genes.

Response: To address this question, we downloaded the nucleotide sequence of the 4,577 ARGs from CARD (version 3.1.4) and performed nucleotide BLAST (blast.ncbi.nlm.nih.gov/Blast.cgi) with our 6 ARGs (*ftsP*, *hdfR*, *lrp*, *proV*, *qorB*, and *rbsK*). However, we did not identify any statistically significant homologs (E-value < 0.05), while the best hit was for OXA-541 of *Pseudomonas putida* for *lrp* (91.7% sequence similarity, E-value = 0.12). Please note that out of the 129 genes from the lowest probability range [0.0,

0.2] that we have validated to have no predicted ARG activity, we found 9 genes that have significant E-value (< 0.05) with >68.6% sequence similarity, arguing that just looking at homology is not sufficient for discovering ARGs. We updated the manuscript to reflect these findings (Results, lines 180-183; Supplementary Information, Section 1.3.12; Supplementary Data 10; Discussion, lines 204-207).

4) Authors did not show how much dissemination these genes have across microbial communities. Such information could help us decided whether these genes are clinically relevant antibiotic resistance genes or not.

Response: Using the MGnify service (<http://www.ebi.ac.uk/metagenomics>), we performed a protein sequence search of our 6 novel ARGs using HMMER (<http://hmmmer.org/>) against their human digestive system microbiome database which contains 94,342 samples. We found how much dissemination these genes have in the database (see table below) for 5 ARGs except for *proV* which kept running into a server-side error (we contacted the support team to have this issue addressed). We updated these results in the manuscript (Results, lines 183-185).

Novel ARG	# of samples containing ARG / Total # of samples	Percentage
lrp	7,292 / 94,342	7.73%
rbsK	8,062 / 94,342	8.55%
qorB	1,963 / 94,342	2.08%
hdfR	8,292 / 94,342	8.79%
ftsP	628 / 94,342	0.67%

5) Authors should also test these genes on different organisms such as *Bacillus subtilis*.

Response: We performed nucleotide mega-BLAST (<blast.ncbi.nlm.nih.gov/Blast.cgi>) to identify the genes homologous to 6 novel ARGs in other bacterial genera. We found that *Salmonella enterica* LT2, a strain from our lab collection, had 5 homologs *ftsP*, *lrp*, *proV*, *rbsK*, and *yifA* (*hdfR* in *E. coli*) with >78% similarity in nucleotide sequences, while the homolog of *qorB* was not identified in *S. enterica*. Wet-lab validation revealed that these 5 homologs also confer resistance to antibiotics. We now include these results as main findings (Abstract, lines 26-27; Introduction, lines 79-80; Results, lines 185-191) as well as details of the wet-lab validation process in the Supplementary Information (Sections 1.3.13 and 1.3.14).

We would like to thank the third reviewer for the valuable feedback!

References

1. Sang, S. *et al.* SemaTyP: a knowledge graph based literature mining method for drug discovery. *BMC Bioinformatics* **19**, 1–11 (2018).
2. Segler, M. & Waller, M. P. Chemical Discovery as a Knowledge Graph Completion Problem. *AITP 2017* (2017).
3. Hassani-Pak, K. & Rawlings, C. Knowledge discovery in biological databases for revealing candidate genes linked to complex phenotypes. *J. Integr. Bioinform.* **14**, (2017).
4. Santos, A. *et al.* Clinical knowledge graph integrates proteomics data into clinical decision-making. *bioRxiv* (2020).
5. Tamae, C. *et al.* Determination of antibiotic hypersensitivity among 4,000 single-gene-knockout mutants of *Escherichia coli*. *J. Bacteriol.* **190**, 5981–5988 (2008).
6. Liu, A. *et al.* Antibiotic sensitivity profiles determined with an *Escherichia coli* gene knockout collection: generating an antibiotic bar code. *Antimicrob. Agents Chemother.* **54**, 1393–1403 (2010).
7. Bordes, A., Usunier, N., Garcia-Duran, A., Weston, J. & Yakhnenko, O. Translating embeddings for modeling multi-relational data. in *Advances in neural information processing systems* 2787–2795 (2013).
8. Ji, G., He, S., Xu, L., Liu, K. & Zhao, J. Knowledge graph embedding via dynamic mapping matrix. in *Proceedings of the 53rd Annual Meeting of the Association for Computational Linguistics and the 7th International Joint Conference on Natural Language Processing (Volume 1: Long Papers)* 687–696 (2015).
9. Balažević, I., Allen, C. & Hospedales, T. M. Tucker: Tensor factorization for knowledge graph completion. *arXiv Prepr. arXiv1901.09590* (2019).
10. Jha, A., Khan, Y., Sahay, R. & d’Aquin, M. Metastatic Site Prediction in Breast Cancer using Omics Knowledge Graph and Pattern Mining with Kirchhoff’s Law Traversal. *bioRxiv* (2020).
11. Chawla, N. V., Bowyer, K. W., Hall, L. O. & Kegelmeyer, W. P. SMOTE: synthetic minority over-sampling technique. *J. Artif. Intell. Res.* **16**, 321–357 (2002).
12. Ernst, P., Siu, A. & Weikum, G. Knowlife: a versatile approach for constructing a large knowledge graph for biomedical sciences. *BMC Bioinformatics* **16**, 157 (2015).
13. Wu, X. *et al.* Automatic knowledge graph construction: A report on the 2019 icdm/icbk contest. in *2019 IEEE International Conference on Data Mining (ICDM)* 1540–1545 (2019).
14. Bosselut, A. *et al.* Comet: Commonsense transformers for automatic knowledge graph construction. *arXiv Prepr. arXiv1906.05317* (2019).
15. Fang, X. *et al.* Global transcriptional regulatory network for *Escherichia coli* robustly connects gene expression to transcription factor activities. *Proc. Natl. Acad. Sci.* **114**, 10286–10291 (2017).
16. Kazemi, S. M. & Poole, D. Simple embedding for link prediction in knowledge graphs. *arXiv Prepr. arXiv1802.04868* (2018).

17. Sun, Z., Deng, Z.-H., Nie, J.-Y. & Tang, J. Rotate: Knowledge graph embedding by relational rotation in complex space. *arXiv Prepr. arXiv1902.10197* (2019).
18. Weiss, S. J., Mansell, T. J., Mortazavi, P., Knight, R. & Gill, R. T. Parallel mapping of antibiotic resistance alleles in *Escherichia coli*. *PLoS One* **11**, e0146916 (2016).
19. Suzuki, S., Horinouchi, T. & Furusawa, C. Prediction of antibiotic resistance by gene expression profiles. *Nat. Commun.* **5**, 5792 (2014).
20. Winkler, J. D. *et al.* The resistome: a comprehensive database of *Escherichia coli* resistance phenotypes. *ACS Synth. Biol.* **5**, 1566–1577 (2016).
21. Yeh, P., Tschumi, A. I. & Kishony, R. Functional classification of drugs by properties of their pairwise interactions. *Nat. Genet.* **38**, 489 (2006).
22. Nichols, R. J. *et al.* Phenotypic landscape of a bacterial cell. *Cell* **144**, 143–156 (2011).

Reviewers' Comments:

Reviewer #1:

Remarks to the Author:

The authors have made substantial improvements that meaningfully strengthen the manuscript. These include:

1. Reducing the scope of the paper's claims to the domain of antibiotic resistance discovery domain which is an accurate and important contribution,
2. Adding multiple graph machine learning baselines (TransE, TransD, TuckER) which demonstrate the value of the authors' approach,
3. Updating the writing throughout for clarity, particularly the methodological details.

In general, KIDS is an interesting work with a nice experimental validation of a computational system for discovering antibiotic resistance genes. The primary value of the paper is the application of knowledge graph methodologies to discover and experimentally validate new genes.

Some final comments:

[Line 105] In general, the authors should soften the claim in the section title "resolved inconsistencies help discover new knowledge." The authors acknowledge that there are a remarkably small set of inconsistencies (236) among the full set of triples in the graph (651,222). They also acknowledge that there are not systematic performance improvements based on this inconsistency resolution although they have a few case studies. While the text is adequately revised elsewhere, it should also be revised here.

[Lines 133-136] There needs to be more detail about the MLP in the methods. As written, it's not clear how the MLP leverages the knowledge graph. It seems just like a fully connected neural network with randomly initialized embeddings for each node in the graph. However, these lines appear to claim that there is some information from the knowledge graph directly that's being used. The MLP strikes me as a non-graph baseline.

[Lines 145-154] The authors claim that adding discovered relationships to their knowledge graph and rerunning their methodology allows the discovery of additional new relationships. Can the authors clarify how much consistency there is in the predictions of the model when it is simply trained multiple times with different seeds? This should be included somewhere in the manuscript as it helps confirm that the new predictions are indeed a result of the added relationships rather than just the stochastic result of retraining the model. Figure 5b-d already make much of this point. Nonetheless, a statement on the model's consistency across training runs would be helpful.

Reviewer #2:

Remarks to the Author:

Thanks for the authors' detailed responses. They addressed most of my concerns. Overall, the reviewer agrees this paper is a nice application of domain knowledge graph construction for a single bacterium, and it shows the inconsistency resolution helps with relation discovery.

The reviewer is curious about the expanded experiments. What is the reason behind the reported predictive performance? For instance, the translational methods in Euclidean space (e.g., TransE and TransD) are not satisfying. Is it because the translational relations are not straightforward? And how the knowledge graph enrichment affects those models? Some intuitive explanation would be helpful to the reader.

One last thing to confirm: do "negative findings" refer to negative instances of triplets during the negative sampling?

Reviewer #3:

Remarks to the Author:

My concerns have been successfully addressed. I strongly recommend the manuscript for publication at Nature communication.

Reviewer #1

The authors have made substantial improvements that meaningfully strengthen the manuscript. These include:

1. Reducing the scope of the paper's claims to the domain of antibiotic resistance discovery domain which is an accurate and important contribution,
2. Adding multiple graph machine learning baselines (TransE, TransD, TuckER) which demonstrate the value of the authors' approach,
3. Updating the writing throughout for clarity, particularly the methodological details.

In general, KIDS is an interesting work with a nice experimental validation of a computational system for discovering antibiotic resistance genes. The primary value of the paper is the application of knowledge graph methodologies to discover and experimentally validate new genes.

Some final comments:

[Line 105] In general, the authors should soften the claim in the section title "resolved inconsistencies help discover new knowledge." The authors acknowledge that there are a remarkably small set of inconsistencies (236) among the full set of triples in the graph (651,222). They also acknowledge that there are not systematic performance improvements based on this inconsistency resolution although they have a few case studies. While the text is adequately revised elsewhere, it should also be revised here.

Response: We agree with the reviewer that the number of inconsistencies is low and that at a systems level, the difference in summary statistics is not significant, something that we discuss in lines 217-222. However, our new results support the header of this section: "Resolved inconsistencies help discover new knowledge." Indeed, after resolution of these inconsistencies, we identified 2 new antibiotic genes, since their score increased by orders of magnitude and were tagged for experimental validation (the only two genes that had this change). The fact that even such a small percentage of inconsistencies lead to the discovery of even 2 new antibiotic genes, demonstrates the need for high quality, clean data sources, which is also the message that we would like to convey here.

[Lines 133-136] There needs to be more detail about the MLP in the methods. As written, it's not clear how the MLP leverages the knowledge graph. It seems just like a fully connected neural network with randomly initialized embeddings for each node in the graph. However, these lines appear to claim that there is some information from the knowledge graph directly that's being used. The MLP strikes me as a non-graph baseline.

Response: The MLP method implicitly learns to place semantically similar entities closer to each other in the embedding space¹ regardless of their proximity in the knowledge graph, and it does not use graph information, although it is considered as a graph-based method in the knowledge graph completion community¹⁻⁴. We agree that these lines "... likely due to exploiting both local (PRA) and global (MLP) patterns" may confuse the readers to think MLP directly extracts features from the knowledge graph. In

this revised manuscript, we have removed the lines above and modified the Results section (lines 128-135) to further clarify these two methods.

[Lines 145-154] The authors claim that adding discovered relationships to their knowledge graph and rerunning their methodology allows the discovery of additional new relationships. Can the authors clarify how much consistency there is in the predictions of the model when it is simply trained multiple times with different seeds? This should be included somewhere in the manuscript as it helps confirm that the new predictions are indeed a result of the added relationships rather than just the stochastic result of retraining the model. Figure 5b-d already makes much of this point. Nonetheless, a statement on the model's consistency across training runs would be helpful.

Response: To address this comment, we tested the consistency of the KIDS-generated hypotheses generated using two metrics Kendall's tau⁵ and rank-biased overlap (RBO)⁶ that checks if two ranked lists are in agreement. Kendall's tau, a widely used correlation-based method, ranges between -1 and +1, where -1 denotes the complete disagreement and +1 denotes the complete agreement between the two ranked lists. However, some of its properties render its application to the KIDS-generated hypotheses less appropriate. For example, Kendall's tau requires two ranked lists to be of the same length, yet the number of hypotheses with probability > 0.20 changes for every run of hypotheses generation due to the stochastic nature of the hypotheses generation models. Moreover, it assigns equal weight to all the items in the ranked list, treating the agreement at the top of the lists (hypotheses with higher probability) as important as the agreement at the bottom of the lists (hypotheses with lower probability). RBO is an alternative method whose value ranges between 0 and 1, where 0 means complete disagreement and 1 means a complete agreement between the two ranked lists. In contrast to Kendall's tau, RBO allows comparing two disjoint ranked lists of different lengths as well as putting more emphasis on the agreement at the top of the lists.

Experimental results show that the KIDS-generated hypotheses are positively correlated with high consistency when compared to the random baseline (Kendall's tau = 0.96 vs. 0.00, respectively, p-value < 2.2×10^{-308} ; RBO = 0.56 vs. 0.00, respectively, p-value < 2.2×10^{-308}), suggesting that the discovery of new relationships in the second iteration are indeed due to the extension of the knowledge graph from the first iteration. Finally, among the 2,907 that appeared at least once among the 100 different versions of first iteration hypotheses with probability > 0.20, we identified 11 hypotheses that appeared in all 100 different versions from which 10 were validated to be a positive relationship (Supplementary Data 11). We now include the methodological details, results, and discussion of this topic in the revised manuscript (Results, lines 156-160; Supplementary Information, Section 1.3.11; Supplementary Data 11).

We appreciate the valuable feedback from reviewer 1.

Reviewer #2

Thanks for the authors' detailed responses. They addressed most of my concerns. Overall, the reviewer agrees this paper is a nice application of domain knowledge graph construction for a single bacterium, and it shows the inconsistency resolution helps with relation discovery.

The reviewer is curious about the expanded experiments. What is the reason behind the reported predictive performance? For instance, the translational methods in Euclidean space (e.g., TransE and TransD) are not satisfying. Is it because the translational relations are not straightforward? And how the knowledge graph enrichment affects those models? Some intuitive explanation would be helpful to the reader.

Response: Although translation-based methods (TransE⁷ and TransD⁸) have shown state-of-the-art performance in benchmark datasets like WN18RR⁹ and FB15k-237¹⁰, they are limited by their inability to handle knowledge graphs with complex and diverse entities and relations (e.g. one-to-many, many-to-one, many-to-many)¹¹ or utilize semantic information³. For example, in our knowledge graph, many *E. coli* genes are known to confer resistance to a specific antibiotic (many-to-one). Therefore, these genes will be close to each other in the embedding space, making it difficult to differentiate them from each other. Although knowledge graph enrichment could potentially be a solution by adding more information to the graph, as these limitations stem from the characteristics of the knowledge graph itself and the nature of the model used, we believe the performance of the hypothesis generators can be improved by taking advantage of the recent language model (LM)-based methods like BERT¹². We now include this discussion in the main manuscript (Discussion, lines 236-253).

One last thing to confirm: do “negative findings” refer to negative instances of triplets during the negative sampling?

Response: ‘Negative findings’ refer to all the negative triples (620,542 triples from 9 triple types) in the knowledge graph that were extracted directly from the source (Supplementary Information, Section 1.1.1) and are treated as ground-truth. These negative findings are different from the negatives generated by randomly corrupting either the subject or the object of the triple, a technique used commonly in link prediction methods to produce negatives for training.

We thank reviewer 2 for the insightful feedback.

Reviewer #3

My concerns have been successfully addressed. I strongly recommend the manuscript for publication at Nature communication.

We thank reviewer 3 for recommending the manuscript for publication.

References

1. Nickel, M., Murphy, K., Tresp, V. & Gabrilovich, E. A review of relational machine learning for knowledge graphs. *Proc. IEEE* **104**, 11–33 (2016).
2. Guan, S., Jin, X., Wang, Y. & Cheng, X. Shared embedding based neural networks for knowledge graph completion. *Symmetry (Basel)*. **13**, 247–256 (2018).
3. Wang, M., Qiu, L. & Wang, X. A Survey on Knowledge Graph Embeddings for Link Prediction. *Symmetry (Basel)*. **13**, 485 (2021).
4. Dong, X. *et al.* Knowledge vault: A web-scale approach to probabilistic knowledge fusion. in *Proceedings of the 20th ACM SIGKDD international conference on Knowledge discovery and data mining* 601–610 (2014).
5. Kendall, M. G. A new measure of rank correlation. *Biometrika* **30**, 81–93 (1938).
6. Webber, W., Moffat, A. & Zobel, J. A similarity measure for indefinite rankings. *ACM Trans. Inf. Syst.* **28**, 1–38 (2010).
7. Bordes, A., Usunier, N., Garcia-Duran, A., Weston, J. & Yakhnenko, O. Translating embeddings for modeling multi-relational data. in *Advances in neural information processing systems* 2787–2795 (2013).
8. Ji, G., He, S., Xu, L., Liu, K. & Zhao, J. Knowledge graph embedding via dynamic mapping matrix. in *Proceedings of the 53rd Annual Meeting of the Association for Computational Linguistics and the 7th International Joint Conference on Natural Language Processing (Volume 1: Long Papers)* 687–696 (2015).
9. Dettmers, T., Minervini, P., Stenetorp, P. & Riedel, S. Convolutional 2d knowledge graph embeddings. in *Thirty-second AAAI conference on artificial intelligence* (2018).
10. Toutanova, K. & Chen, D. Observed versus latent features for knowledge base and text inference. in *Proceedings of the 3rd Workshop on Continuous Vector Space Models and their Compositionality* 57–66 (2015).
11. Feng, J. *et al.* Knowledge graph embedding by flexible translation. in *Fifteenth International Conference on the Principles of Knowledge Representation and Reasoning* (2016).
12. Devlin, J., Chang, M.-W., Lee, K. & Toutanova, K. Bert: Pre-training of deep bidirectional transformers for language understanding. *arXiv Prepr. arXiv1810.04805* (2018).